# Oscillatory cAMP signaling rapidly alters H3K4 methylation

Tyler C Huff[1],*, Vladimir Camarena[1],*, David W Sant[1], Zachary Wilkes[1], Derek Van Booven[1], Allegra T Aron[2], Ryan K Muir[3], Adam R Renslo[3], Christopher J Chang[2,4,5], Paula V Monje[6], Gaofeng Wang[1,7]

Epigenetic variation reflects the impact of a dynamic environment on chromatin. However, it remains elusive how environmental factors influence epigenetic events. Here, we show that G protein–coupled receptors (GPCRs) alter H3K4 methylation via oscillatory intracellular cAMP. Activation of Gs-coupled receptors caused a rapid decrease of H3K4me3 by elevating cAMP, whereas stimulation of Gi-coupled receptors increased H3K4me3 by diminishing cAMP. H3K4me3 gradually recovered towards baseline levels after the removal of GPCR ligands, indicating that H3K4me3 oscillates in tandem with GPCR activation. cAMP increased intracellular labile Fe(II), the cofactor for histone demethylases, through a non-canonical cAMP target—Rap guanine nucleotide exchange factor-2 (RapGEF2), which subsequently enhanced endosome acidification and Fe(II) release from the endosome via vacuolar H$^+$-ATPase assembly. Removing Fe(III) from the media blocked intracellular Fe(II) elevation after stimulation of Gs-coupled receptors. Iron chelators and inhibition of KDM5 demethylases abolished cAMP-mediated H3K4me3 demethylation. Taken together, these results suggest a novel function of cAMP signaling in modulating histone demethylation through labile Fe(II).

## Introduction

Cellular systems constantly respond to a barrage of environmental stimuli by transducing extracellular signals into transcriptional changes. G protein–coupled receptors (GPCRs) are the largest and most diverse group of membrane receptors which sense extracellular changes by binding with specific ligands (Lefkowitz, 2007). The binding of agonists to Gs-coupled receptors elevates, whereas binding to Gi-coupled receptors suppresses, the second messenger cAMP to induce downstream molecular changes in response to environmental stimuli (Sutherland, 1970; Sunahara et al, 1996). Under physiological conditions, stimuli for GPCRs are often persistent and periodic which could result in a long-term oscillation of intracellular cAMP (Dyachok et al, 2006). Furthermore, activators or inhibitors of adenylate cyclases (ACs), which produce cAMP, and of phosphodiesterase (PDE), which degrade cAMP, can directly change the level of intracellular cAMP. For example, bicarbonate and caffeine both increase intracellular cAMP by activating soluble AC and inhibiting PDE, respectively. The signal transduction of GPCRs via cAMP has been extensively studied for decades and is thought to be well established. The impact of cAMP on gene transcription is considered to be mediated by three transcription factors (CREB, ATF1, and CRE) which can be phosphorylated by cAMP-dependent PKA (Montminy, 1997). The phosphorylation of these transcription factors generally activates gene expression and is thought to be the primary link between cAMP signaling and transcription (Sands & Palmer, 2008).

We recently reported that cAMP also influences transcription by promoting DNA hydroxymethylation, the initial step of active DNA demethylation (Camarena et al, 2017). This effect was found to be mediated by a cAMP-induced elevation of intracellular labile Fe(II), an essential cofactor for ten-eleven translocation (TET) methylcytosine dioxygenases responsible for DNA demethylation. TETs belong to the Fe(II) and 2-oxoglutarate (2OG, alternatively termed α-ketoglutarate)–dependent dioxygenase superfamily. Without Fe(II), the reaction catalyzed by these dioxygenases would be halted (Tahiliani et al, 2009). However, Fe(II) is tightly controlled in the cell largely because of its ability to produce free radicals through the Fenton reaction (Dunn et al, 2007). We showed that elevation of intracellular cAMP increases the intracellular labile Fe(II) pool, which further enhances DNA hydroxymethylation and changes the transcriptome (Camarena et al, 2017). Thus, environmental factors, by stimulating Gs-/Gi-coupled receptors or by directly affecting the

---

[1]John P. Hussman Institute for Human Genomics, Dr. John T. Macdonald Foundation Department of Human Genetics, University of Miami Miller School of Medicine, Miami, FL, USA   [2]Department of Molecular and Cell Biology, University of California, Berkeley, CA, USA   [3]Department of Pharmaceutical Chemistry, University of California, San Francisco, CA, USA   [4]Department of Chemistry, University of California, Berkeley, CA, USA   [5]Howard Hughes Medical Institute, University of California, Berkeley, CA, USA   [6]Department of Neurological Surgery, Indiana University School of Medicine, Indianapolis, IN, USA   [7]Sylvester Comprehensive Cancer Center, University of Miami Miller School of Medicine, Miami, FL, USA

Correspondence: gwang@med.miami.edu
*Tyler C Huff and Vladimir Camarena contributed equally to this work

activity of AC/PDE, could alter the intracellular labile Fe(II) pool, DNA methylation, and gene transcription via the second messenger cAMP.

JmjC domain–containing histone demethylases, such as TETs, also belong to the Fe(II) and 2OG–dependent dioxygenase superfamily, indicating that labile Fe(II) is essential for JmjC-mediated histone demethylation (Tsukada et al, 2006). This raises a possibility that cAMP signaling might also regulate histone demethylation. Here, we report that activation of Gs-coupled receptors caused a rapid loss of histone methylation, especially H3K4me3, an effect that was mimicked by cAMP analogues and forskolin but was blocked by AC inhibition. Conversely, stimulation of Gi-coupled receptors quickly elevated H3K4me3, which is inhibited by forskolin. The effect of cAMP signaling on H3K4me3 was mediated by labile Fe(II) and was blocked by iron chelators. In the absence of Fe(III) in the medium, activation of Gs-coupled receptors no longer augmented the intracellular labile Fe(II) pool. Knockout of Rap guanine nucleotide exchange factor-2 (RapGEF2) abolished the effect of cAMP signaling on vacuolar H$^+$-ATPase assembly, endosome acidification, and subsequent intracellular labile Fe(II) elevation. Upon ligand removal, H3K4me3 gradually recovers towards baseline levels. Collectively, this study may provide insight into the regulation of histone demethylation by cAMP signaling, which could be implicated in human health and disease.

# Results

## cAMP rapidly and specifically reduces H3K4 methylation

We previously reported that intracellular cAMP elevation induces DNA demethylation in a variety of cell types by augmenting the intracellular labile Fe(II) pool (Camarena et al, 2017). Since JmjC domain-containing histone demethylases require Fe(II) as an essential cofactor, we speculated if cAMP could also alter histone demethylation. To test this hypothesis, we treated cultured Schwann cells with membrane-permeable cAMP analogue 8-CPT-cAMP (hereafter denoted as cAMP) and assessed levels of four well-characterized histone 3 (H3) trimethylation marks: H3K4me3, H3K9me3, H3K27me3, and H3K36me3. Prolonged cAMP treatment (8–48 h) led to a reduction in H3K4me3, but not other trimethylation marks (Fig 1A and B). A comparable change in H3K4me3 was also observed using immunofluorescence (IF) (Fig S1). Our existing RNA-seq data revealed that the top three highly expressed JmjC domain-containing histone demethylases in Schwann cells are KDM5A, KDM5B, and KDM5C, which preferentially demethylate H3K4. In comparison, the level of KDM5 expression is 3.2–47.5-fold of other KDMs (Table S1), which may explain at least partially that cAMP

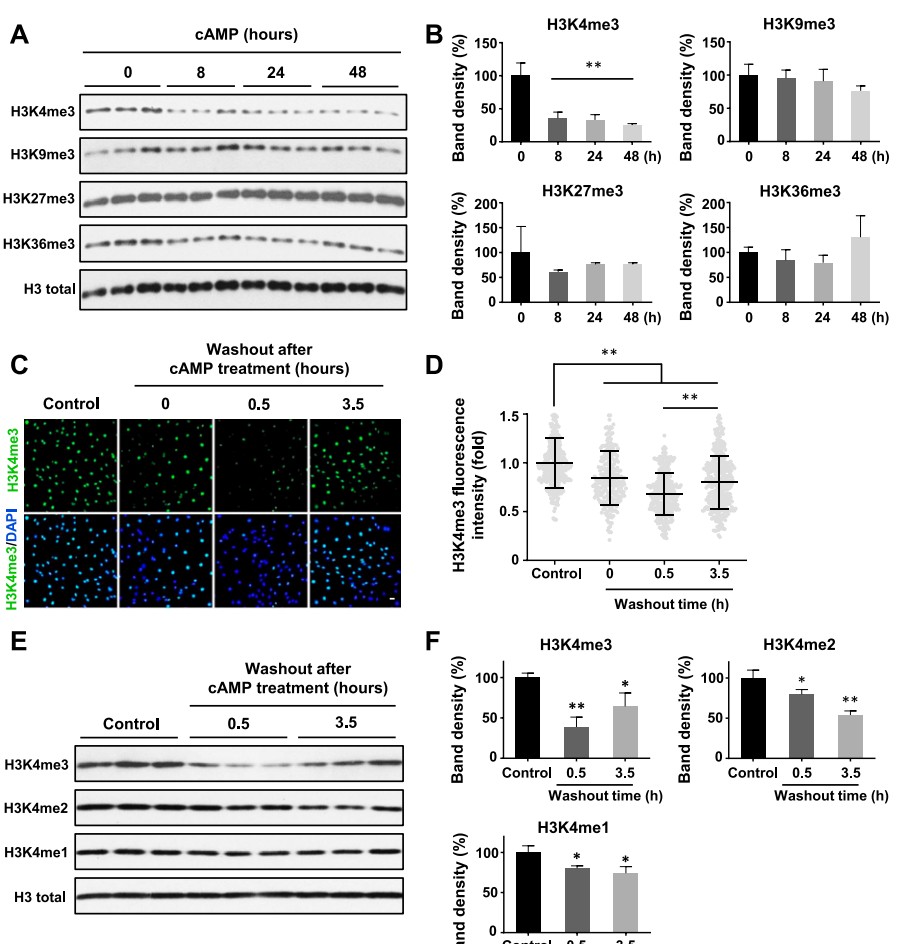

**Figure 1. cAMP decreases H3K4 trimethylation.**
**(A)** Immunoblot of histone trimethylation marks in Schwann cells treated with membrane-permeable 8-CPT-cAMP (100 μM) for various times. **(B)** Band density quantifications show that cAMP treatment reduces H3K4me3 but not H3K9me3, H3K27me3, or H3K36me3 in Schwann cells (n = 3). **(C)** IF of H3K4me3 after cAMP (50 μM) treatment followed by washout. **(D)** Quantification of IF shows that H3K4me3 is markedly decreased 30 min after washout, which rebounds towards baseline levels after 3 h 30 min (n > 250 data points per condition). **(E)** Immunoblot of H3K4 methylation marks in Schwann cells after cAMP (50 μM) treatment followed by washout. **(F)** Band density quantifications show that unlike H3K4me3, which decreases then recovers towards baseline levels, H3K4me2 and H3K4me1 are reduced but without obvious recovery towards the baseline after 3 h 30 min washout after treatment (n = 3). *$P < 0.5$, **$P < 0.01$. Statistical differences were determined by one-way ANOVA. All data are means ± SD. Scale bar = 20 μm.

signaling causes a significant decrease in H3K4me3 in Schwann cells compared with other methylated histone marks.

Intracellular cAMP levels vary according to cell type, ranging from 10 nM to 10–50 µM after stimulation of Gs-coupled receptors (Conti et al, 2014). Because the cellular permeability of 8-CPT-cAMP is ~20% (Werner et al, 2011), treatment with 100 µM 8-CPT-cAMP can increase intracellular cAMP to ~20 µM, which is within the endogenous range. We next tested if alterations in endogenous cAMP affect H3K4me3. Treatment with AC activator forskolin and PDE inhibitors caffeine and 3-isobutyl-1-methylxanthine (IBMX), all agents that increase intracellular cAMP, caused rapid loss of H3K4me3 (Fig S2). These results suggest that cAMP specifically reduces global H3K4me3 in Schwann cells and that continuous cAMP treatment maintains this status.

We then focused on H3K4me3 by first testing how quickly H3K4me3 was altered upon cAMP treatment. cAMP induced maximal H3K4me3 demethylation as early as 30 min of continuous treatment (Fig S3). Intracellular cAMP often oscillates under physiological conditions (Dyachok et al, 2006). To mimic this condition, the cells were briefly treated with cAMP which was subsequently removed by washout and medium change. Although short (30 min) treatment with cAMP reduced H3K4me3, levels of H3K4me3 appeared to rebound back towards baseline 3 h 30 min after washout (Fig 1C and D). We also tested A2058 melanoma cells which, like Schwann cells, are of neural crest origin and may express similar JmjC domain–containing histone demethylases and, therefore, exhibit H3K4me3 demethylation upon cAMP treatment. The rapid reduction in H3K4me3, but not other trimethylation marks, was also observed in A2058 cells along with a swift recovery towards baseline levels after cAMP removal (Fig S4). Collectively, these results suggest that cAMP quickly triggers a global reduction in H3K4me3, which is not limited to Schwann cells but is likely a general effect of cells that express similar demethylase profiles.

To assess global changes in H3K4 methylation after cAMP stimulation, we then measured H3K4me2 and H3K4me1. Comparatively, both H3K4me2 and H3K4me1 only slightly decreased upon cAMP treatment. Unlike H3K4me3, after removal of cAMP for 3 h 30 min, H3K4me2 and H3K4me1 levels remained low without obvious recovery towards baseline levels (Fig 1E and F). These data show that methylation at H3K4, especially H3K4me3, fluctuates rapidly and dynamically in response to cAMP.

### Gs-/Gi-coupled receptors regulate H3K4me3

The signaling of Gs- or Gi-coupled receptors is mainly mediated by cAMP, the second messenger. These receptors either increase intracellular cAMP by activating AC or decrease intracellular cAMP by inhibiting AC. We reasoned that because H3K4me3 decreases upon elevation of intracellular cAMP after treatment with membrane-permeable cAMP, AC activators, and PDE inhibitors, then perhaps GPCRs may also regulate H3K4me3. We first tested Gs-coupled calcitonin gene-related peptide (CGRP) receptors that have been shown to elevate intracellular cAMP in Schwann cells (Cheng et al, 1995). We found that stimulation with CGRP dose-dependently decreased H3K4me3 (Fig S5). Brief stimulation with CGRP (30 min) followed by washout caused a reduction in H3K4me3 30 min after washout (Fig 2A and C). Similar to exogenous cAMP treatment, H3K4me3 levels began to rebound towards the baseline after

washout for 3 h 30 min. The reduction of H3K4me3 was also observed after stimulation of β-adrenergic receptor, another Gs-coupled receptor expressed in Schwann cells (Fig S6). To test if cAMP indeed mediated the effect of Gs-coupled receptors on H3K4me3, an AC inhibitor was applied before agonist stimulation. Treatment with AC inhibitor SQ22536 abolished the reduction of H3K4me3 induced by CGRP or isoproterenol and H3K4me3 levels rebounded even higher than the baseline after ligand removal for 3 h 30 min (Figs 2A and C, and S6). These results suggest that Gs-coupled receptors, via cAMP, rapidly reduce H3K4me3.

If Gs-coupled receptors reduce H3K4me3 by up-regulating intracellular cAMP, Gi-coupled receptors could conversely increase H3K4me3 by down-regulating intracellular cAMP. We then tested Gi-coupled lysophosphatidic acid receptors ($LPAR_1$ and $LPAR_3$), which have been shown to diminish intracellular cAMP in Schwann cells (Anliker et al, 2013). LPA treatment (1 h) caused an increase in H3K4me3 after washout for 1 h. The level of H3K4me3 consequently declined towards the baseline after washout for 3 h (Fig 2B and D). Furthermore, pretreatment with the AC activator forskolin hindered the effect of LPA on H3K4me3 (Fig 2B and D), suggesting that Gi-coupled receptors, via cAMP, quickly increase H3K4me3. Taken together, these data suggest that Gs-/Gi-coupled receptors regulate H3K4me3.

### Rapid labile Fe(II) elevation underpins cAMP-induced H3K4me3 demethylation

JmjC-mediated histone demethylation requires the following components: the enzyme JmjC domain–containing histone demethylases (various kinds expressed in most cells), the substrate-methylated lysine in histones (abundant in the chromatin), the cofactor Fe(II), the co-substrate 2OG (abundant in healthy cells), and oxygen (accessible to cultured cells). Of all these components, Fe(II) is tightly controlled in the cell because of its ability to produce free radicals through the Fenton reaction (Dunn et al, 2007). Given the observation that H3K4me3 is rapidly altered by cAMP signaling (within 1 h), new protein synthesis is unlikely to participate in this regulation. Our earlier work shows that cAMP promotes TET-mediated DNA demethylation by elevating labile Fe(II) (Camarena et al, 2017), which was measured by Trx-Puro probes (Spangler et al, 2016) after cAMP analogue or GPCR ligand treatment for more than 3 h. It is plausible that the effect of cAMP on histone demethylation is also mediated by labile Fe(II). However, it is unclear if cAMP can induce a rapid change in labile Fe(II).

We used a newly developed FIP-1 probe to measure intracellular labile Fe(II) after cAMP stimulation. This probe is a highly sensitive Fe(II) sensor which uses fluorescence resonance energy transfer (FRET) to measure changes in labile Fe(II) in real time using microscopy (Aron et al, 2016). cAMP stimulation for 30 min induced a near 20% peak increase in labile Fe(II) (Fig 3A and B). Levels of labile Fe(II) began to decline to the baseline after washout for 1 h. The transient elevation of labile Fe(II) caused by brief cAMP treatment is correlated with H3K4me3 reduction induced by the same treatment. Furthermore, LPA treatment, via Gi-coupled $LPAR_1$ and $LPAR_3$, diminished intracellular labile Fe(II) (Fig S7), which is also associated with H3K4me3 up-regulation. These results indicate a dynamic relationship between levels of cAMP and the intracellular labile Fe(II) pool.

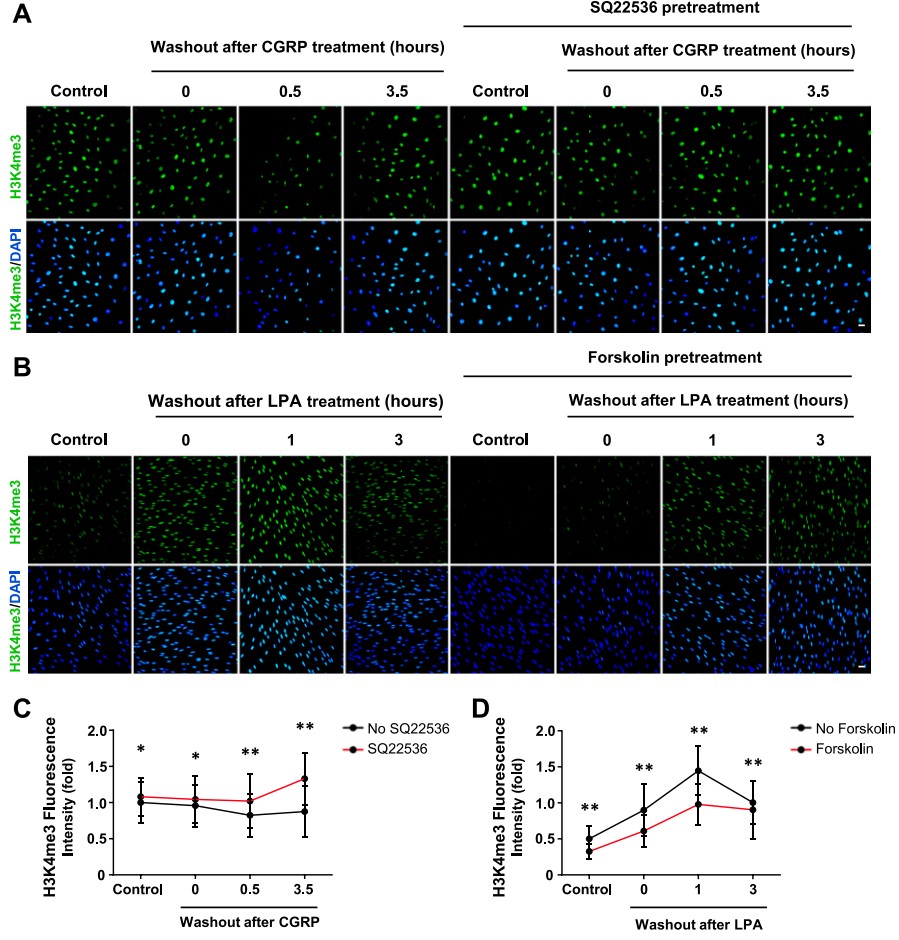

**A**

SQ22536 pretreatment

Washout after CGRP treatment (hours)  |  Washout after CGRP treatment (hours)

Control  0  0.5  3.5  |  Control  0  0.5  3.5

H3K4me3

H3K4me3/DAPI

**B**

Forskolin pretreatment

Washout after LPA treatment (hours)  |  Washout after LPA treatment (hours)

Control  0  1  3  |  Control  0  1  3

H3K4me3

H3K4me3/DAPI

**C**

H3K4me3 Fluorescence Intensity (fold)

Control  0  0.5  3.5
Washout after CGRP

— No SQ22536
— SQ22536

**D**

H3K4me3 Fluorescence Intensity (fold)

Control  0  1  3
Washout after LPA

— No Forskolin
— Forskolin

**Figure 2. GPCR stimulation alters H3K4 trimethylation.**
**(A)** IF of H3K4me3 after CGRP (10 nM) treatment for 30 min followed by washout, with or without pretreatment of AC inhibitor SQ22536 (100 $\mu$M). **(B)** IF of H3K4me3 after LPA (2 $\mu$M) treatment for 1 h followed by washout, with or without pretreatment of AC activator forskolin (10 $\mu$M). **(C)** Quantification of IF shows that brief stimulation with $G_s$-coupled receptor ligand CGRP followed by washout induces peak demethylation of H3K4me3 at 30 min after washout. Pretreatment with SQ22536 abolishes CGRP-induced H3K4me3 reduction (n > 250 data points per condition). **(D)** Quantification of IF shows that stimulation with $G_i$-coupled receptor ligand LPA (2 $\mu$M) followed by washout promotes massive H3K4 trimethylation 1 h after washout which retreats towards baseline levels after 3 h. Pretreatment with forskolin abrogates LPA-induced H3K4 trimethylation 1 h after washout (n > 250 data points per condition). *$P < 0.5$, **$P < 0.01$. Statistical differences were determined by two-way ANOVA. All data are means ± SD. Scale bar = 20 $\mu$m.

We then tested whether elevated labile Fe(II) mediates the effect of cAMP on H3K4me3. Fe(II), which is scarce in the extracellular milieu in vivo, can directly enter into the cell through divalent metal cation transporter 1 (DMT1) (De Domenico et al, 2008). The addition of Fe(II) to the cell media resulted in a time-dependent decrease in H3K4me3 levels (Fig 3C and D). Conversely, the addition of iron chelators 2,2 bipyridyl and deforexamine before cAMP stimulation abolished the effect of cAMP on H3K4me3 (Fig 3E). Overall, these data suggest that cAMP rapidly elevates the intracellular labile Fe(II) pool, which consequently induces H3K4me3 demethylation.

The canonical cAMP signaling involves PKA, cyclic nucleotide-gated channels (CNGCs), and exchange protein directly activated by cAMP (Epac) (Gloerich & Bos, 2010). To assess whether these canonical targets of cAMP underlie cAMP-mediated demethylation of H3K4me3, we pretreated cells with the Epac inhibitor ES109, PKA inhibitors KT5720 and H89, or CNGC blocker l-cis-diltiazem. Pretreatment with any of these inhibitors and blocker did not sufficiently block cAMP-mediated H3K4me3 reduction (Fig S8). This is consistent with our previous report that cAMP elevates labile Fe(II) via a non-canonical pathway. Collectively, our data suggest that cAMP induces H3K4me3 demethylation by augmenting the intracellular labile Fe(II) pool and that this phenomenon occurs independently of the canonical cAMP signaling pathway.

**cAMP signaling elevates labile Fe(II) via RapGEF2-dependent V-ATPase assembly**

Intracellular labile Fe(II) is largely governed by Fe(III) uptake from the extracellular space and subsequent reduction to Fe(II) in the endosome. We, therefore, tested whether extracellular Fe(III) was required for cAMP-induced intracellular Fe(II) elevation. In the absence of Fe(III) in the media, isoproterenol, a ligand for Gs-coupled $\beta$-adrenergic receptors which increase intracellular cAMP, no longer elevated intracellular labile Fe(II) in Schwann cells (Fig 4A), suggesting that cellular iron uptake is indeed involved in the up-regulation of Fe(II) by cAMP signaling.

Fe(III) is internalized via transferrin–transferrin receptor-mediated endocytosis, which localizes to the endosome. There, endosome acidification by the vacuolar $H^+$-ATPase (V-ATPase) causes release of Fe(III) from transferrin and is subsequently converted to Fe(II) (Nishi & Forgac, 2002). We previously showed that cAMP signaling enhances endosome acidification and that silencing of RapGEF2, a non-canonical target of cAMP, abolishes the effect of cAMP on endosome acidification and labile Fe(II) (Camarena et al, 2017). However, it remains largely unclear how cAMP signaling affects the acidification of endosomes. It is plausible that cAMP signaling may induce endosome acidification by affecting the function or the

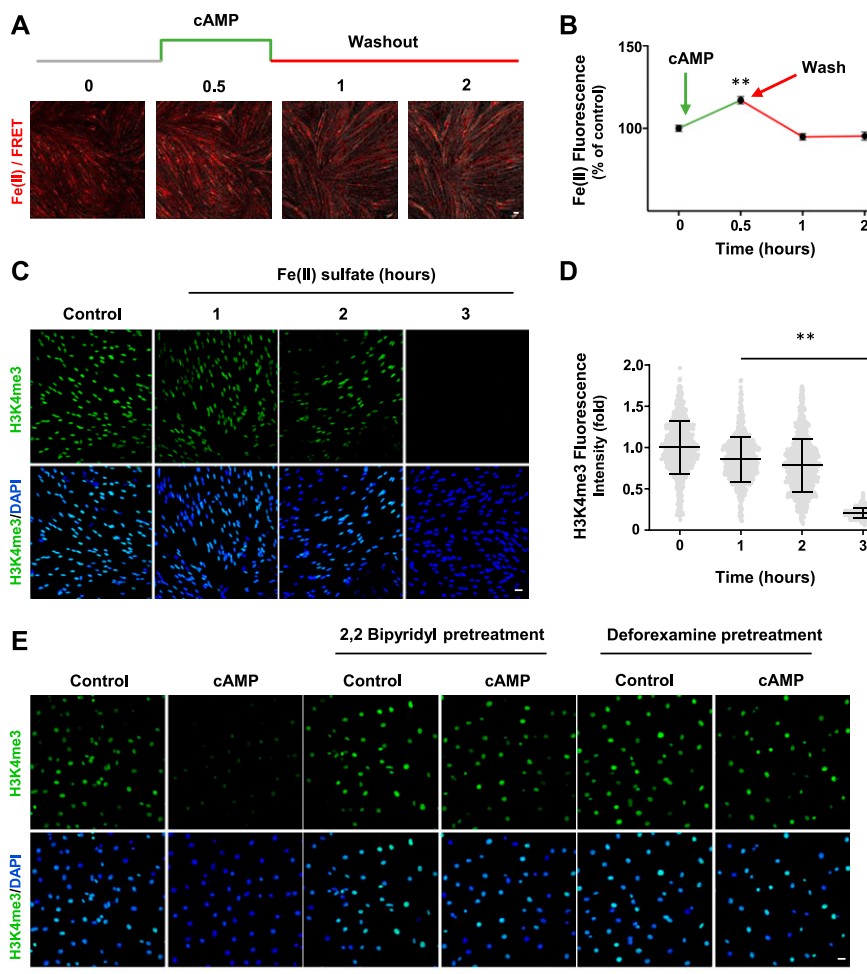

**Figure 3. Intracellular labile Fe(II) mediates cAMP-induced H3K4me3 demethylation.**
**(A)** Live imaging of Schwann cells pretreated with the Fe(II)-sensitive FIP-1 probe and briefly treated with cAMP (100 μM) followed by washout. Images represent the ratio of the Fe(II) channel/FRET channel (pseudo-colored red/yellow). Increased red/yellow signal indicates an increase in intracellular Fe(II). **(B)** Fluorescence quantification shows that cAMP treatment induces a peak increase of labile Fe(II) 30 min after washout which dissipates within 1 h (n > 250 data points per condition). **(C)** IF of H3K4me3 after Fe(II) sulfate (0.1 μM) treatment for various times. **(D)** Quantification of IF shows that Fe(II) sulfate decreases H3K4me3 in a time-dependent manner (n > 250 data points per condition). **(E)** Pretreatment with iron chelators 2,2 bipyridyl and deforexamine (20 μM) abolishes cAMP-induced H3K4me3 demethylation. *$P < 0.5$, **$P < 0.01$. Statistical differences were determined by one-way ANOVA. All data are means ± SD. Scale bar = 20 μm.

assembly of V-ATPase. We tested if cAMP signaling changes V-ATPase assembly by assessing V-ATPase subunit $V_1A$ abundance in endosome-enriched cellular fractions as a measure of assembly and $V_0D$ as a loading control (McGuire & Forgac, 2018). RapGEF2 knockout neuroscreen-1 (NS1-RapGEF2 KO) cells and the wild-type NS-1 (NS1-WT) control cells were used in this experiment (Emery et al, 2013; Jiang et al, 2017). We found that forskolin treatment induced massive up-regulation of V-ATPase subunit $V_1A$ in the membrane fraction of NS1-WT cells, indicating increased V-ATPase assembly. The enhanced V-ATPase assembly was verified in HEK-293 cells after treatment with isoproterenol (Fig S9). However, forskolin-induced V-ATPase assembly was attenuated in RapGEF2 knockout NS1 cells (NS1-RapGEF2 KO) cells (Fig 4B). These results suggest that cAMP signaling promotes V-ATPase assembly and is likely dependent on RapGEF2.

By ablation of cAMP-induced V-ATPase assembly, knockout of RapGEF2 could further impair endosome acidification. Indeed, we found that deletion of RapGEF2 abolished forskolin-induced endocytotic vesicle acidification compared with the wild-type cells (NS1-WT) (Fig 4C). It is known that Fe(II) release from the endosome into the intracellular labile Fe(II) pool relies on endosome acidification. In wild-type cells, forskolin treatment up-regulated labile

Fe(II). In contrast, the same treatment failed to increase labile Fe(II) in RapGEF2 knockout cells (Fig 4D). Similarly, treatment with Gs-coupled ligands isoproterenol or CGRP induces up-regulation of labile Fe(II) in NS1-WT cells, whereas this effect was abolished in cells lacking RapGEF2 (Figs 4E and FigsS10). Collectively, these data suggest that cAMP induces RapGEF2-dependent V-ATPase assembly which consequently up-regulates endosome acidification and labile Fe(II)—the cofactor for JmjC domain–containing histone demethylases.

## Rapid H3K4me3 demethylation and subsequent recovery are mediated by KDM5 and histone methyltransferases (HMTs)

The dynamics of H3K4me3 are governed by the activities of HMTs and JmjC domain–containing demethylases. Again, we probed our existing RNA-seq data and found that the top three highest expressed JmjC domain–containing histone demethylases in Schwann cells are KDM5A, KDM5B, and KDM5C, which preferentially demethylate H3K4 (Table S1) (Brier et al, 2017). It could be that KDM5 is responsible for the rapid decrease of H3K4me3 caused by cAMP signaling. To test this hypothesis, we pretreated Schwann cells with

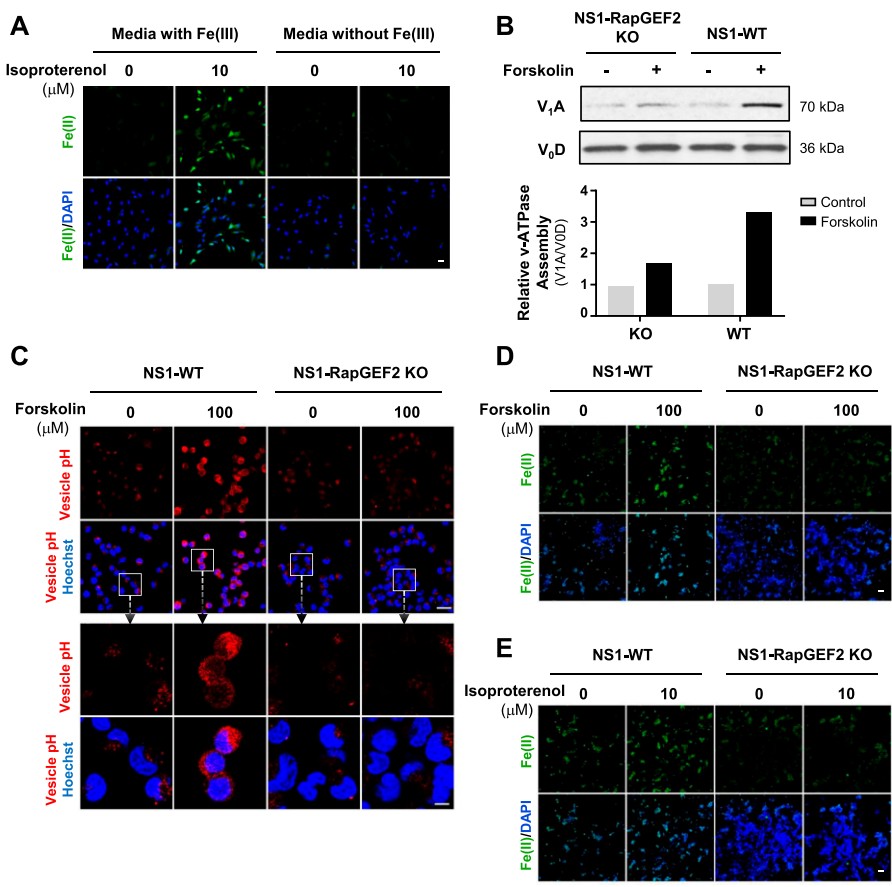

**Figure 4. cAMP-induced Fe(II) elevation is likely mediated by RAPGEF2 and endosome acidification. (A)** IF of labile Fe(II) using TRX-Puro ferrous iron probe in Schwann cells cultured in media containing Fe(III) or media without Fe(III) then treated with Gs-coupled receptor ligand isoproterenol (10 $\mu M$). Scale bar = 20 $\mu m$. **(B)** Immunoblot of V-ATPase subunits $V_1A$ and $V_0D$ in wild-type NS1 cells (NS1-WT) and RapGEF2 knockout cells (NS1-RapGEF2 KO) treated with forskolin (100 $\mu M$). Band density quantifications of the relative ratio between $V_1A$ and $V_0D$ show that forskolin treatment promotes vATPase assembly in NS1-WT cells but is ablated in NS1-RapGEF2 KO cells. **(C)** Fluorescence-based vesicular pH probe signal after forskolin (100 $\mu M$) treatment in NS1-WT and NS1-RAPGEF2 KO cells. Red fluorescence increases when the pH decreases from 8 to 4. Scale bar = 40 $\mu m$. Boxes enclose high magnification images of cells and endocytotic vesicles. Scale bar = 100 $\mu m$. **(D)** IF of labile Fe(II) using TRX-Puro ferrous iron probe in NS1-WT and RAPGEF2 KO cells after forskolin (100 $\mu M$) treatment. Scale bar = 10 $\mu m$. **(E)** IF of labile Fe(II) using TRX-Puro ferrous iron probe in NS1 WT and RAPGEF2 KO cells after treatment with isoproterenol. Scale bar = 10 $\mu m$.

a pan KDM5 inhibitor KDOAM-25. Without KDOAM-25 pretreatment, short-term cAMP treatment causes a rapid reduction in H3K4me3 at 30 min after washout which rebounds towards baseline levels after 3 h 30 min. In comparison, pretreatment with KDOAM-25 completely abolished the effect of cAMP on H3K4me3 (Fig 5A and C). We then used siRNA silencing of KDM5A/B/C isoforms to further validate their role in cAMP-induced H3K4me3 demethylation. cAMP caused significant H3K4me3 demethylation in Schwann cells, provided no siRNA or treated with non-targeting (Scramble) siRNA. However, treatment with KDM5 siRNA abolished cAMP-induced H3K4me3 demethylation (Fig S11). Considering that the expression of KDM5 remains relatively unchanged after cAMP treatment (Camarena et al, 2017), these results suggest that upon brief cAMP signaling and subsequent transient labile Fe(II) up-regulation, the catalytic activity of KDM5 is enhanced and H3K4me3 is subsequently demethylated.

To test whether HMTs underlie the rebound of H3K4me3 after cAMP washout, we pretreated cells with a cocktail consisting of HMT inhibitors MI-2 and MM-102 before briefly treating cells with cAMP and following up with washout. Without HMT inhibitors, H3K4me3 rebounds towards baseline levels 3 h 30 min after washout (Fig 5B and D). In contrast, pretreatment with HMT inhibitors abolishes the rebound of H3K4me3 and prolongs cAMP-induced H3K4me3 demethylation after washout (Fig 5B and D), suggesting that HMTs are responsible for H3K4me3 recovery after cAMP-induced H3K4me3 demethylation. Overall, upon activation

of cAMP signaling, KDM5 and HMTs work cooperatively in tandem to dynamically regulate H3K4me3, which could consequently alter transcription.

### cAMP induces H3K4me3 demethylation mostly within promoter regions

To further understand the impact of cAMP signaling on H3K4me3 in the chromatin, we performed ChIP-seq on Schwann cells briefly treated with cAMP followed by washout. Consistent with the immunoblot results, cAMP treatment induces a widespread reduction of H3K4me3 peaks 30 min after washout followed by a restoration of H3K4me3 after 3 h 30 min (Fig 6A). Differential peak analysis showed that 1,391 peaks were down-regulated and 856 peaks were up-regulated at 30 min, resulting in an observed net decrease in H3K4me3. Although a trend was observed, differential peak analysis found that 2,364 peaks were down-regulated and 2,277 peaks were up-regulated at 3 h 30 min, resulting in no net increase at 3 h 30 min likely because of the high variability between samples in this condition. About 50% of all H3K4me3 peaks are down-regulated 30 min after washout, which are restored after 3 h 30 min (Table S2). Interestingly, more than 90% of all called peaks are in the same locations across all three time points (Fig 6B), suggesting that the major changes after cAMP stimulation occur at the same loci. Furthermore, H3K4me3 was found to be highly enriched in gene promoter regions with more than 50% of all H3K4me3 promoter

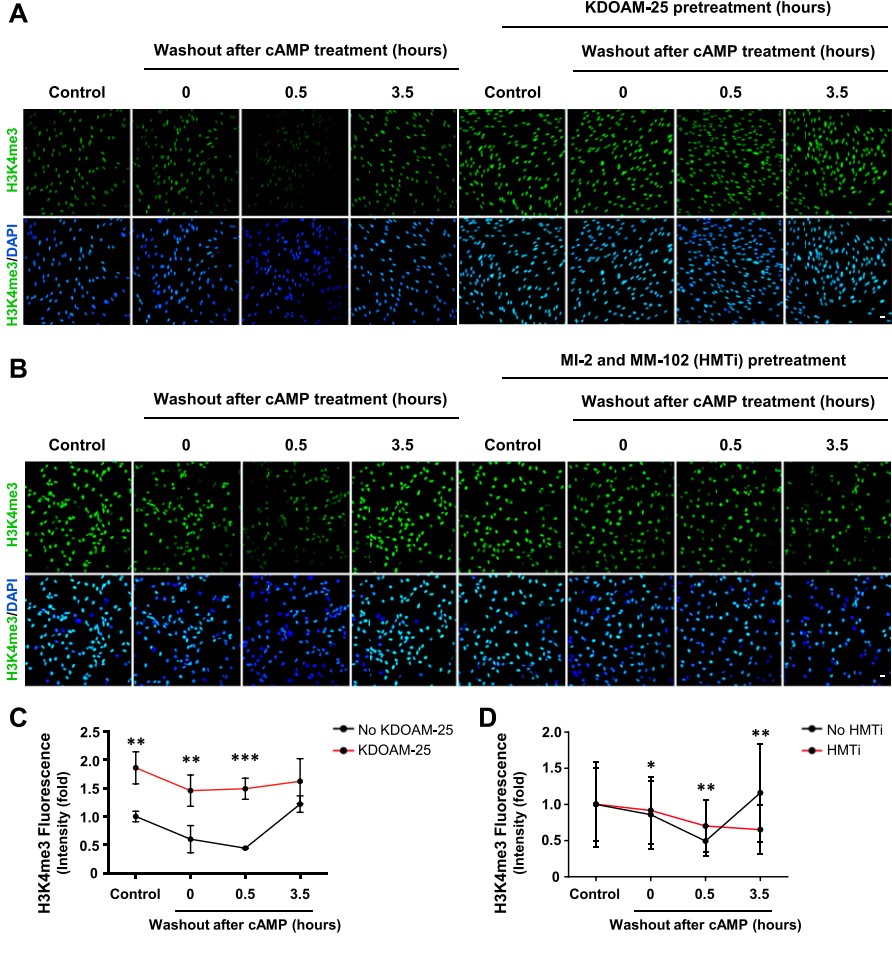

**Figure 5. KDM5 demethylases and histone methyltransferases underlie H3K4me3 dynamics after cAMP treatment.**
**(A)** IF of H3K4me3 after 8-CPT-cAMP (50 µM) treatment for 30 min followed by washout, with or without pretreatment of KDM5 inhibitor KDOAM-25 (100 µM). **(B)** IF of H3K4me3 after 8-CPT-cAMP (50 µM) treatment for 30 min followed by washout, with or without pretreatment of HMT inhibitors MI-2 (20 µM) and MM-102 (20 µM). **(C)** Quantification of IF shows that brief treatment with cAMP followed by washout induces peak demethylation of H3K4me3 30 min after washout. Pretreatment with pan KDM5 demethylase inhibitor KDOAM-25 abolishes the effect of cAMP on H3K4me3 (n > 250 data points per condition). **(D)** Quantification of IF shows that pretreatment with HMT inhibitors MI-2 and MM-102 blocks the restoration of H3K4me3 after 3 h 30 min and slightly hinders cAMP-induced H3K4me3 demethylation at 30 min after washout (n > 250 data points per condition). *P < 0.5, **P < 0.01. Statistical differences were determined by two-way ANOVA. All data are means ± SD. Scale bar = 20 µm.

peaks down-regulated 30 min after washout which subsequently return to baseline levels after 3 h 30 min (Fig 6C and Table S3). This effect was seen genome-wide in a diverse set of genes, including *Samd1*, *Fpgt*, and many others (Fig 6D and Tables S4, and S5). Overall, our data reveal a novel cAMP signaling pathway which connects periodic GPCR signaling with rapid changes in histone methylation (Fig 7).

## Discussion

Posttranslational modification of histones by methylation is dynamic and reversible, which plays critical roles in development and disease (Greer & Shi, 2012). It is known that certain environment factors, such as dietary consumption of methionine—the precursor of methyl donors—could change the status of histone methylation (Mentch et al, 2015). However, besides methionine metabolism, it remains elusive if and how numerous extracellular molecules affect histone methylation. Membrane receptors such as GPCRs, the largest and most diverse receptor family, sense environmental cues by binding with ligands (Sutherland, 1970). The binding of GPCRs with corresponding ligands triggers intracellular signaling. cAMP is the second messenger of numerous Gs- or Gi-coupled receptors,

which upon stimulation increase or decrease intracellular cAMP, respectively. Under physiological conditions, the agonists for GPCRs are often persistent and periodic. Consequently, the level of intracellular cAMP oscillates, like waves (Dyachok et al, 2006). In the current study, we show that transient elevation of intracellular cAMP, by brief stimulation of Gs-coupled receptors or short exposure to membrane-permeable cAMP, causes wave-like changes in H3K4me3—a rapid reduction followed by recovery towards baseline levels. Comparatively, transient reduction of intracellular cAMP by quick stimulation of Gi-coupled receptors initiates an opposite, yet similar wave-like change in H3K4me3—a rapid elevation followed by recovery towards baseline levels. Collectively, our data suggest that extracellular GPCR stimulation via cAMP oscillates with H3K4me3 in the chromatin.

The rapid fluctuation of H3K4me3 by GPCRs appears to be mediated by labile Fe(II), an essential cofactor for JmjC domain–containing demethylases. JmjC domain–containing demethylases contain more than 20 members and can remove methyl group from tri-, di-, and monomethylated histone lysine residues (Justin et al, 2010). Without any stimulation, baseline H3K4me3 remains relatively stable. When Fe(II) is added to the media and enters the intracellular labile Fe(II) pool, methylation at H3K4 especially H3K4me3 is rapidly decreased, suggesting that intracellular labile

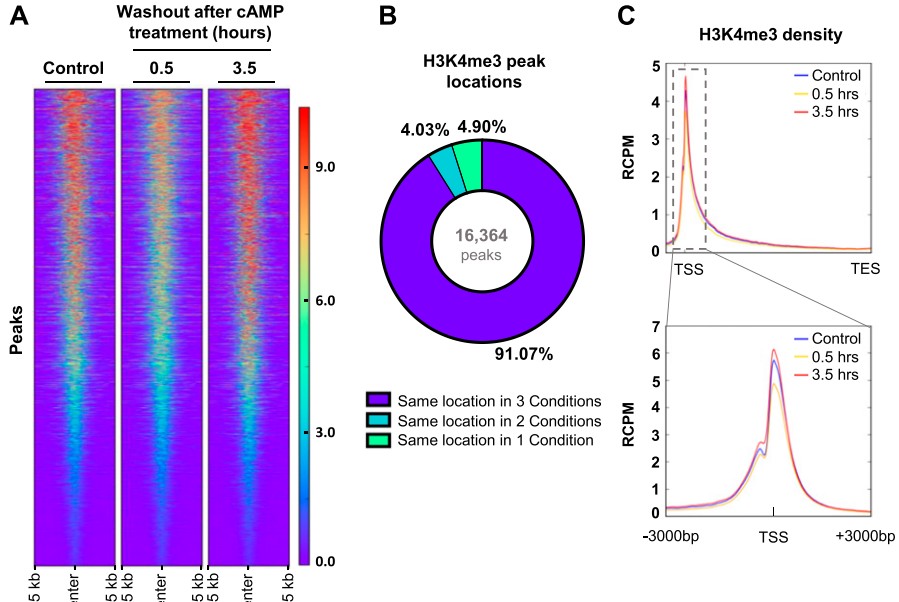

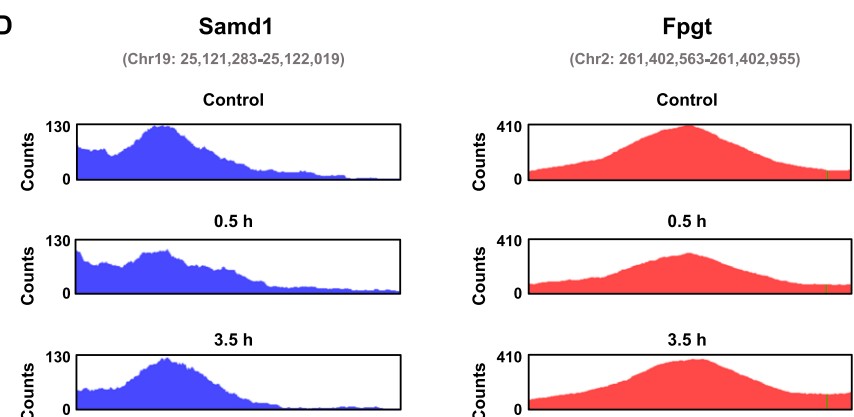

**Figure 6. ChIP sequencing reveals that cAMP induces rapid, widespread changes in H3K4me3.**
**(A)** Heat map of all detected H3K4me3 peaks after brief cAMP (50 $\mu M$) treatment followed by washout. **(B)** Chart representing the percentage of peaks called in one condition, two conditions, or all conditions. A vast majority of peaks were called from the same location in every condition. **(C)** Brief treatment with cAMP (50 $\mu M$) followed by washout induces peak demethylation of H3K4me3 at promoter regions 30 min after washout. **(D)** Representative H3K4me3 peaks from *Samd1* and *Fpgt* which exhibited H3K4me3 demethylation 30 min after washout and recovery of H3K4me3 after 3 h 30 min. Plots were created using Integrative Genomics Viewer. Peak coordinates are indicated in gray. n = 3 for ChIP sequencing experiment.

Fe(II) promotes the catalytic activity of KDM5, which is highly expressed in Schwann cells and antagonizes H3K4 methylation. Our earlier work showed that cAMP signaling regulates the intracellular labile Fe(II) pool through a previously unrecognized pathway involving the noncanonical cAMP target RapGEF2, Rap1 activation, endosome acidification, and subsequent labile Fe(II) release from the endosome (Camarena et al, 2017). We now show that cAMP signaling up-regulates labile Fe(II) via iron cellular uptake. Our results further demonstrate that cAMP enhances the assembly of the V-ATPase, which is responsible for pumping protons into endosomes. Using RapGEF2 null cells, we consolidate that RapGEF2 is essential in the up-regulation of endosome acidification and subsequent labile Fe(II) release by cAMP signaling. It is, thus, plausible that cAMP targets RapGEF2, which activates Rap1 and in turn promotes V-ATPase assembly. This warrants further detailed studies in the future.

Treatment with cAMP or Fe(II) caused widespread demethylation of H3K4me3 but not other trimethylated histone marks. This could be due to a relatively higher expression of KDM5, which antagonizes

H3K4 methylation, in Schwann cells. Our existing RNA-seq data revealed that the highest expressed JmjC domain–containing histone demethylases in Schwann cells are KDM5A, KDM5B, and KDM5C, with a combined expression that is 3.2–47.5-fold higher than other KDMs (Table S1). Increased expression of KDM5 could, therefore, explain widespread H3K4me3 demethylation in response to Fe(II) elevation. In addition, H3K4me3 has the highest turnover rate of any trimethylation mark with a half-life of 6 h 48 min (Zheng et al, 2014). It has been posited that this could be due to either (1) an increased basal activity of JmjC domain–containing demethylases which target H3K4me3 or (2) transcription-dependent eviction of nucleosomes near the transcription start site which are enriched in H3K4me3. Furthermore, restriction of methionine, the methyl donor for methyltransferases, in both cells and mice causes a drastic reduction in H3K4me3 compared with other trimethylated histone marks (Mentch et al, 2015; Dai et al, 2018). Although future studies are warranted to elucidate mechanistic processes underlying H3K4me3 turnover, these observations suggest that H3K4me3 is relatively unstable and may, therefore, be more sensitive to

**Gs-coupled receptor activation**

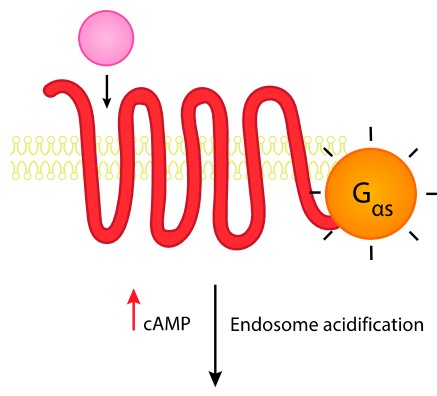

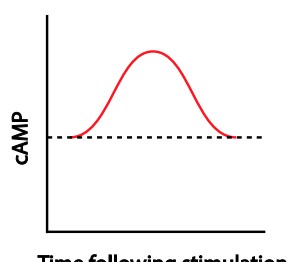

**Figure 7. Working model of cAMP-induced H3K4me3 demethylation.**
Activation of Gs-coupled receptors leads to a transient increase in cAMP, the second messenger, which results in RapGEF-mediated V-ATPase assembly and endosome acidification. Acidification causes release of Fe(III) from transferrin–transferrin receptor complex and is converted to Fe(II) by Steap3 before leaving the endosome and elevating the labile Fe(II) pool. Increased intracellular labile Fe(II) leads to increased activity of KDM5 demethylases and results in transient demethylation of H3K4me3.

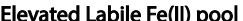

↑cAMP    Endosome acidification

**Elevated Labile Fe(II) pool**

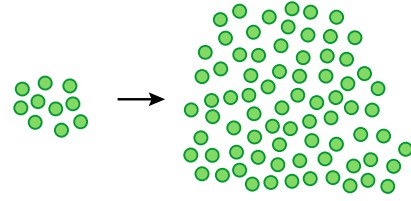

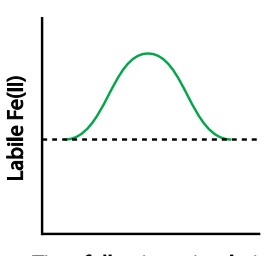

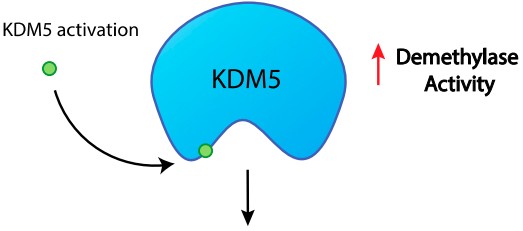

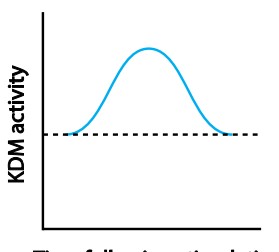

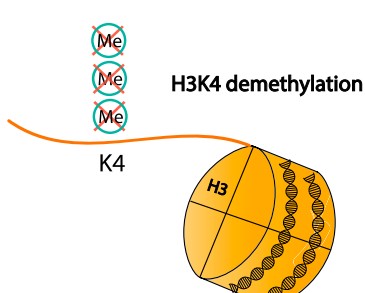

**H3K4 demethylation**

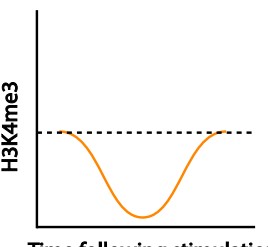

demethylation upon cAMP-induced Fe(II) elevation compared with other histone marks.

Continuous treatment with cAMP or stimulation of Gs-coupled receptors elevates intracellular labile Fe(II). To mimic the physiological oscillation of intracellular cAMP, we treated cells in this study briefly with cAMP which induced a relatively transient increase of intracellular labile Fe(II) followed by a decline towards the baseline after washout. The rapid wave-like change in the labile Fe(II) pool correlates inversely with the wave form pattern in H3K4me3 after cAMP treatment and washout. When iron chelators are applied, intracellular cAMP elevation no longer induces any changes in H3K4me3. Thus, periodic activation of GPCRs could constantly change H3K4me3 through oscillations in intracellular cAMP, concurrent fluctuations in labile Fe(II), and subsequent changes in KDM5 activity. Cells are constantly exposed to a barrage of extracellular stimuli and must dynamically respond to them to

function and survive. cAMP-induced histone demethylation via GPCR activation could be a signaling mechanism used by the cell to swiftly alter transcription in response to a dynamically changing microenvironment. Using cAMP, the evolutionarily conserved second messenger, repurposes an intracellular process that is not only rapid but also responsive to a diversity of extracellular stimuli which activate GPCRs, thus conferring a more nuanced cellular response to dynamic environmental challenges.

The dynamic fluctuation of histone lysine methylation modulates chromatin structure, genome stability, and ultimately gene transcription (Klose et al, 2006). Our study indicates that this process is also coordinated in response to cAMP signaling. Histone methylation at different lysine residues leads to diverse consequences in transcription. H3K4me3, especially when located in the promoter region, is associated with actively transcribed genes (Santos-Rosa et al, 2002; Guenther et al, 2007; Heintzman et al, 2007). By ChIP-seq, we confirmed that genome-wide H3K4me3 decreases upon treatment with cAMP and recovers towards the baseline after cAMP removal. Changes in H3K4me3 peaks detected by ChIP-seq were relatively modest compared with results obtained via IF or Western blot for overall H3K4me3. It is not exactly clear to us why only modest changes were discovered in the ChIP-seq analysis. We found that very high percentage of reads reside within large peaks, which may consequently affect the read coverage in other relatively smaller peaks that could be altered by the treatment. This read coverage bias towards large peaks may explain the relatively modest changes by ChIP-seq analysis. Although the fold changes detected in the genomic analysis are modest, they were found to be statistically significant and may nonetheless be biologically relevant to transcription. H3K4me3 is known to be co-transcriptionally deposited (Woo et al, 2017). Although H3K4me3 is classically associated with the promoters of transcriptionally active genes, it remains unclear whether cAMP-induced H3K4me3 is a consequence or a determinant of the differentially transcribed genes. It is known that cAMP has an impact on transcription (Montminy, 1997). Currently, it is believed that PKA-dependent phosphorylation of the transcription factors cAMP response element-binding protein (CREB), cAMP response element modulator (CREM), and activating transcription factor (ATF) are principally responsible for changes in gene expression after cAMP signaling. However, our data show an alternative mechanism, whereby cAMP signaling induces swift changes in labile Fe(II) to alter KDM activity and subsequently cause rapid changes in H3K4me3, which could potentially alter transcription independently of PKA and other canonical cAMP targets. Furthermore, we previously reported that cAMP signaling could affect the transcriptome by DNA demethylation (Camarena et al, 2017). Now it is likely that GPCRs, via cAMP signaling, may regulate gene transcription through three mechanisms: PKA-dependent transcription factors, DNA methylation, and histone methylation. Future studies are warranted to dissect how cAMP signaling coordinates these three mechanisms in gene transcription.

By regulating the intracellular labile Fe(II) pool, GPCRs via cAMP affect demethylation of both DNA and histones. Because Fe(II) is tightly controlled within the cell (Dunn et al, 2007), there could be a competition between TET-mediated DNA demethylation and JmjC-mediated histone demethylation. It is unclear if and how histone demethylation interacts with DNA demethylation under the influence of cAMP signaling. In this study, we mainly examined primary cultured Schwann cells which are not plagued by mutations as in most cell lines. Furthermore, these cells grow slowly which allows us to examine the impact of GPCRs on histone methylation without frequent cell cycle interference. The regulation of histone demethylation could be a general effect of cAMP signaling, as the suppression of H3K4me3 by cAMP was also verified in A2058 melanoma cells. A2058 cells, like Schwann cells and melanocytes, are of neural crest origin and may share similar JmjC domain–containing histone demethylase expression profiles, which could explain why H3K4me3 is preferentially demethylated in both cell types. Because all JmjC domain–containing histone demethylases use Fe(II) for demethylase activity, it is expected that cAMP signaling would fluctuate different histone marks depending on the JmjC histone demethylase expression profile of a given cell type. Furthermore, nearly 50% of all marketed drugs target various GPCRs, many of which are transduced through cAMP signaling. However, the impact of these drugs on the demethylation of DNA and histones has been overlooked. Other agents such as caffeine in beverages and bicarbonate, the baking soda, could change the epigenome by directly elevating intracellular cAMP. These drugs and agents may exert a long-lasting impact on human health, disease, and even inheritance through epigenetic mechanisms which are just now being elucidated.

In conclusion, GPCRs regulate JmjC-mediated histone demethylation through cAMP signaling and intracellular labile Fe(II). This previously unrecognized function of cAMP signaling could have wide-reaching implications in human health and disease.

## Materials and Methods

### Cell culture

Primary cultured Schwann cells were isolated from 3-mo-old Fisher rat sciatic nerves as previously described (Monje, 2018). Briefly, nerves were cut into small segments, incubated in vitro for 10 d in DMEM medium containing 10% heat-inactivated FBS and allowed to degenerate. Degenerated nerve explants were dissociated using a 0.25% dispase/0.05% collagenase solution and the resulting cell suspension was plated on poly-L-lysine-coated dishes (Sigma-Aldrich). Purified primary Schwann cells were expanded up to passage one in DMEM media containing 10 nM neuregulin (recombinant heregulin-$\beta$1) and 2 $\mu$M forskolin. Experiments were performed on Schwann cells between three and six passages and plated on glass coated with poly-L-lysine and laminin from Engelbreth-Holm-Swarm murine sarcoma basement membrane. Schwann cells were grown for experiments in DMEM supplemented with 10% FBS and without neuregulin or forskolin.

For cAMP washout experiments, Schwann cells were treated with 8-CPT-cAMP (8-(4-chlorophenylthio) adenosine-3′,5′-cyclic monophosphate) for 30 min, washed quickly with PBS, and then given new DMEM media. The cells were collected at the following time points after washout: 0 (30 min of treatment, collect without washing), 30 min (30 min of treatment, wash, replace media, and collect after 30 min), and 3 h 30 min (30 min of treatment, wash, replace media, and

collect after 3 h 30 min). For experiments involving inhibitors, all cells were pretreated and resupplied with the inhibitor following washout.

## Immunoblot

After treatment, the cells were washed with PBS and lysed with radioimmunoprecipitation assay (RIPA) buffer buffer containing phosphatase and protease inhibitors. Cell lysates were resuspended in Laemmli SDS sample buffer (60 mM Tris–HCl, 1% SDS, 10% glycerol, 0.05% bromophenol blue, pH 6.8, and 2% $\beta$-mercaptoethanol) and then subjected to 10% SDS–PAGE. The gels were then transferred to polyvinylidene difluoride (PVDF) membrane for immunoblot. Membranes were then blocked in 5% non-fat skim milk for 1 h and then incubated with primary antibody at 4° overnight. Primary antibodies directed against H3K4me3 (Cat. no. 39915; Active Motif), H3K4me2 (Cat. no. 9725; Cell Signaling), H3K4me1 (Cat. no. 9723; Cell Signaling), H3K9me3 (Cat. no. 39161; Active Motif), H3K27me3 (Cat. no. 39156; Active Motif), H3K36me3 (Cat. no. 61101; Active Motif), and H3 total (Cat. no. 39763; Active Motif) were used. Membranes were then washed three times before incubating with peroxidase-conjugated secondary antibody. Protein bands were detected using a chemiluminescence kit (Millipore).

## IF

Cells were seeded onto 24-well plates containing 12-mm glass coverslips at 10,000–50,000 cells per well. After treatments were performed, the cells were fixed with 4% PFA, blocked with 5% bovine serum albumin, and 0.4% Triton X-100 in PBS for 1 h, then incubated with anti-H3K4me3 primary antibody (Cat. no. 39915; Active Motif, 1:1,000) at 4° overnight. The cells were then incubated with Alexa Fluor 488–conjugated donkey antirabbit IgG (1:1,000) and counterstained with DAPI.

## Image acquisition and analysis

Images from IF experiments were captured using a Zeiss LSM 710 confocal microscope at a bit depth of 8 and into a 512 × 512 frame size. Fluorescence intensity was quantified using Fiji (ImageJ) (Schindelin et al, 2012). Pixel intensity values were averaged 16 times and measured from every cell within the image field from a minimum of five fields per condition (~450 cells total). The intensity values from every cell within the image field were plotted and analyzed by one- and two-way ANOVA with Tukey's post hoc test and Bonferroni correction using GraphPad Prism 7 from GraphPad Software.

## Intracellular labile Fe(II) detection

Live cell imaging of labile Fe(II) was measured by FIP-1 probe as previously described (Aron et al, 2016). Briefly, the cells were washed with HBSS (containing calcium and magnesium) and incubated with 10 $\mu$M FIP-1 in HBSS for 1 h. The cells were washed three times and maintained in HBSS at 37°C, and live images were acquired for 30 min every 5 min using a Zeiss laser confocal microscope 710. The cells were treated with 8-CPT-cAMP (100 $\mu$M) and another 30 min of images were acquired. Media was then washed three times with HBSS and live images were acquired for

an additional 1 h 30 min. FIP-1 was excited using a 488-nm laser ("Green" channel and "FRET" channel). "Green" emission was collected using a META detector between 500 and 535 nm, whereas "FRET" emission was collected using a META detector between 555 and 611 nm. The ratio of the green channel/FRET channel was considered as the signal and was pseudo-colored red/yellow. Analysis and quantification were performed using ImageJ. Statistical analyses for multiple comparisons were carried out through one-way ANOVA with the Bonferroni correction using the software R. Labile Fe(II) was also detected using TRX-Puro probes as described in our previous studies (Spangler et al, 2016; Camarena et al, 2017).

## Vesicular pH detection

Endocytotic vesicle pH was assessed by pHrodo Red Dextran (Thermo Fisher Scientific) according to the manufacturer's protocol. Briefly, the cells were washed three times with PBS after cAMP or forskolin treatment and incubated with pHrodo Red Dextran (20 $\mu$g/ml) for 20 min in the incubator at 37°. The cells were then washed again and fixed with 4% PFA for 10 min. After fixation, the cells were counterstained with Hoechst 33342 or DAPI. Images were captured by a Zeiss LSM 710 confocal microscopy.

## V-ATPase assembly

This method was performed as previously described (Shao & Forgac, 2004) with small modifications. Briefly, 1 × 10^7 cells were plated in 10-cm plates. After treatment, the cells were taken to a 4°C cold room and washed twice with ice-cold PBS before collecting in 650 $\mu$l of homogenization buffer (250 mM sucrose, 1 mM EDTA, 10 mM Hepes, 1 mM PMSF, protease inhibitor cocktail [Sigma-Aldrich], and phosphatase inhibitor cocktail [Thermo Fisher Scientific]). The cells were lysed by passing with them through a 28G syringe five times. Nuclei and non-fragmented cells were precipitated by centrifuging at 500$g$ for 10 min. The membrane fraction was precipitated by ultracentrifugation of the supernatant at 50,000$g$ for 2 h. The cytosolic fraction in the remaining supernatant was concentrated using Amicon Ultra 10k centrifugal filters. The membrane and cytosolic fractions were resuspended with RIPA buffer (50 mM Tris–HCl, 150 mM NaCl, 0.1% SDS, 0.5% sodium deoxycholate, and 1% NP40) plus 2% SDS and protease inhibitors and homogenized with three cycles (30 s On/OFF at 4°C in high) in a Bioruptor sonitcator (Diagenode). Protein concentration was determined using the BCA assay (Thermo Fisher Scientific). Before SDS–PAGE, cell lysates were resuspended in SDS sample buffer (60 mM Tris–HCl, 1% SDS, 10% glycerol, 0.05% bromophenol blue, pH 6.8, with 2% $\beta$-mercaptoethanol). Samples were subjected to 10% SDS–PAGE (Bio-Rad) and transferred to PVDF membranes (Bio-Rad) for immunoblot. Transfer efficiency was determined by Ponceau S staining (Sigma-Aldrich). PVDF membranes were incubated with blocking solution (TBS containing 0.1% Tween 20% and 5% milk) and were probed with specific antibodies including V_1A (H00000523-A01; Abnova, 1:1,000) and V_0D (ab56441; Abcam, 1:1,000). Protein bands were detected using a chemiluminescence kit (Millipore). Band intensity was quantified using Fiji (ImageJ). Relative V-ATPase assembly was determined by

measuring the ratio between the subunits $V_1A$ and $V_0D$ in the membrane and normalized to the ratio found in control samples.

### KDM5 siRNA gene silencing

Small short interfering RNA (siRNA) was used to transiently silence *KDM5A*, *KDM5B*, and *KDM5C* in Schwann cells during cAMP experiments. To decrease the expression of KDM5 isoforms, Accell siRNA duplexes targeting *KDM5A* (Cat. no. A-095654-24-0010; Dharmacon), *KDM5B* (Cat. no. A-082318-16-0010; Dharmacon), *KDM5C* (Cat. no. A-095923-14-0010; Dharmacon), and non-targeting Scramble control siRNA (Cat. no. D-001910-01-05; Dharmacon) were used. siRNAs combined and delivered in DMEM containing 2.5% FBS for a final concentration of 1 $\mu$M according to the manufacturer's instructions. After two 72-h transfections, the cells were treated with cAMP and collected after 8 h to assess H3K4me3 via IF.

### ChIP-seq

Schwann cells were cultured in 150-mm plates until confluency and chromatin was collected using the ChIP-iT High Sensitivity Kit (Cat. no. 53040; Active Motif) as instructed by the manufacturer. Briefly, the cells were fixed with 1% formaldehyde for 15 min, collected via scraping, and then sonicated using a Bioruptor Pico (Diagenode). DNA was sheared to 200-bp fragments and chromatin-containing H3K4me3 was immunoprecipitated overnight at 4° while rocking. Antibody was pulled down using Protein G beads and separated via chromatography. Chromatin samples were de-crosslinked and DNA was purified using the provided purification columns. Enriched DNA and the unprecipitated genomic input controls were prepared for sequencing using the NEBNext Adaptor Ligation kit (New England Biolabs Inc.). Input and ChIP samples were sequenced on a HiSeq-3000 single-read 75-bp flow cell.

Reads were trimmed with trim_galore to remove low-quality bases from reads (scores <20 in Phred+33 format), and Illumina adapters. Sequence reads were aligned to the rat genome (Rnor_6.0, Ensembl.org) using Burrows-Wheeler Alignment tool (Li & Durbin, 2009). Multi-mapped reads were removed using SAMtools and duplicate reads were removed using PicardTools (https://broadinstitute.github.io/picard/) (Li et al, 2009). Peaks were called and filtered using the Irreproducible Discovery Rate method developed for the ENCODE project (Li et al, 2011). Briefly, peak calling was performed with MACS2 using the narrow peak mode and a relaxed threshold of 0.001 (Zhang et al, 2008). Peaks between treatments were merged and a total of 16,364 peaks were investigated further. Reads within peak regions were quantified using HT-Seq-count and differential enrichment was calculated using edgeR (Robinson et al, 2010; Anders et al, 2015).

### Statistical analysis

Quantitative data are presented as the mean and SD. GraphPad Prism was also used to analyze data and generate graphs. *t* test and one-way *ANOVA* were used to compare two or more groups, respectively, and $P < 0.05$ was considered as statistically significant.

## Data Availability

ChIP sequencing data have been deposited in Gene Expression Omnibus (accession: GSE125728).

## Supplementary Information

## Acknowledgements

This work is supported by National Institutes of Health (NIH) grants (R01NS089525 to G Wang and GM079465 to CJ Chang) and a Craig H Neilsen Foundation grant (#339576 to PV Monje). G Wang is supported by Sylvester NIH Funding Program from Sylvester Comprehensive Cancer Center at the University of Miami. CJ Chang is an Investigator with the Howard Hughes Medical Institute and a Canadian Institute for Advanced Research Senior Fellow. AT Aron thanks the NSF for a graduate fellowship and was partially supported by a Chemical Biology Training Grant from the NIH (T32 GM066698). The authors would like to thank the National Institute of Mental Health for providing NS1 WT and NS1 RapGEF2 KO cells.

### Author Contributions

TC Huff: formal analysis, investigation, methodology, and writing—original draft, review, and editing.
V Camarena: formal analysis, investigation, methodology, and writing—review and editing.
DW Sant: data curation, formal analysis, investigation, and visualization.
Z Wilkes: formal analysis.
D Van Booven: formal analysis and visualization.
AT Aron: resources.
RK Muir: resources.
AR Renslo: resources.
CJ Chang: resources.
PV Monje: resources, supervision, funding acquisition, and methodology.
G Wang: conceptualization, funding acquisition, project administration, and writing—review and editing.

### Conflict of Interest Statement

The authors declare that they have no conflict of interest.

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
