## [Reviewer comments · Life Science Alliance]

Oscillatory cAMP Signaling Rapidly Alters H3K4 Methylation

Tyler Huff, Vladimir Camarena, David Sant, Zachary Wilkes, Derek Van Booven, Allegra Aron, Ryan Muir, Adam Renslo, Christopher Chang, Paula Monje, and Gaofeng Wang
DOI: <https://doi.org/10.26508/lsa.201900529>

Corresponding author(s): Dr. Gaofeng Wang (University of Miami)

Review timeline:

Submission Date:	2019-08-20
Editorial Decision:	2019-09-18
Revision Received:	2019-11-11
Editorial Decision:	2019-12-08
Revision Received:	2019-12-10
Accepted:	2019-12-16

Scientific Editor: Andrea Leibfried

Transaction Report:

1st Editorial Decision

18 September 2019

September 18, 2019

Re: Life Science Alliance manuscript #LSA-2019-00529

Prof. Gaofeng Wang
University of Miami
1501 NW 10th Avenue
Miami 33136

Dear Dr. Wang,

Thank you for submitting your manuscript entitled "Oscillatory cAMP Signaling Rapidly Alters H3K4 Methylation and Gene Expression" to Life Science Alliance. The manuscript was assessed by expert reviewers, whose comments are appended to this letter.

As you will see, the reviewers find your work of interest, but think that the directness/specificity of the effects of cAMP on H3K4me3 level changes and transcription needs to get further analyzed. Given the input provided by the reviewers, we would like to invite you to submit a revised version of your manuscript to us, addressing the concerns raised. Importantly, in addition to testing for direct effects (rev#1-3), cell proliferation and cell cycle analyses need to get included (rev#1) and the inhibitor-based studies need further validation (rev#1). Statistical analyses should get performed as well (rev#3).

Thank you for this interesting contribution to Life Science Alliance. We are looking forward to receiving your revised manuscript.

Sincerely,

Andrea Leibfried, PhD
Executive Editor

Life Science Alliance
Meyershofstr. 1
69117 Heidelberg, Germany
t +49 6221 8891 502
e a.leibfried@life-science-alliance.org
www.life-science-alliance.org

B. MANUSCRIPT ORGANIZATION AND FORMATTING:

Reviewer #1 (Comments to the Authors (Required)):

This manuscript reports that cAMP regulates H3K4 methylation levels. These effects are shown to involve changes in intracellular Fe(II) levels, a cofactor for histone demethylases. Using mainly inhibitors followed by immunofluorescence detection of H3K4me3, it is shown that Fe(II) and KDM5 demethylase were required for the observed decrease in H3K4me3 levels following cAMP addition. The changes in intracellular Fe(II) by cAMP appears to involve RAPGEF2 and endosome acidification. ChIP-Seq analysis confirmed TSS associated changes in H3K4me3 levels and RNA-Seq analyses showed that these changes accompanied gene expression dysregulation. This study identifies a link between cAMP signaling, Fe(II) regulation and gene expression through H3K4me3. These results are interesting and for the most

part the data are rigorously analyzed. However, this manuscript is rather descriptive and relies almost entirely on inhibitors and IF. Some additional supporting evidence for the proposed model should be provided before publication.

Main issue

1. How does cAMP addition affect cell proliferation and cycling? These may contribute to these results on histone methylation changes so this analysis should be performed. In Fig 1E and F, why are H3K4me3 levels increased after washout while diMe2 isn't, which is another substrate for KDM5 demethylases? Could there be additional effects on other enzymes that could explain these results?
2. This manuscript relies almost exclusively on inhibitors. Some genetic experiments should be performed to validate these results. When genetic experiments were performed in Fig 4, H3K4me3 and Fe(II) levels were not even examined. Fig 4 appears to be a little peripheral to the main story so the experiments performed on H3K4me3 should be done for Fig 4D and 4E (4E lacks labels for the IF images as well).
3. It is surprising that there are no changes in H3K4me3 levels in control and time 0 experiments when histone demethylase and methyltransferase inhibitors are used. How do the authors know that these are working? Similarly, why is there no difference between control and 0 hr in Fig 2C while clearly in B there is a large difference in fluorescence intensity?
4. What type of genes are dysregulated by cAMP treatment in Fig 6? Are they related to cAMP and GPCR signaling? For the most highly altered H3K4me3 peaks and gene expression alterations, it would be nice to see some representative examples showing ChIP-Seq and RNA-Seq data.

Reviewer #2 (Comments to the Authors (Required)):

Huff et al. use cAMP pathway inhibitors and activators to show that the cAMP signaling transiently alters the levels of H3K4 trimethylation in cultured Schwann cells, and the kinetics of this effect requires histone methyltransferases and histone demethylases. Furthermore, intracellular Fe(II) is necessary and sufficient for the observed effect, likely involving RapGEF2. Finally, the authors performed ChIP-seq and nascent RNA-seq experiments after 0.5 h cAMP treatment and a short washout (0.5 h) or a long washout (3.5 h). They find that promoters with high levels H3K4me3 are downregulated globally after the short washout and back to normal after the long washout. The long washout timepoint is accompanied by an increase in nascent transcript levels relative to the short washout. They conclude that cAMP affects both H3K4me3 levels and gene expression.

The main finding, that cAMP levels can globally regulate H3K4me3 is interesting since it makes a novel connection between stress, metabolism and global regulation of gene expression. The experiments in this manuscript clearly support the role of cAMP signaling in modulating the levels of H3K4Me3 in Schwann cells. However, how the changes in H3K4me3 levels affect the transcription of genes is not well addressed and the corresponding analysis of the nascent transcript levels is questionable. Therefore, analysis of these data and the conclusions need to be addressed before publication. The manuscript could also benefit from rewriting and spelling out the larger implications of the finding.

Major points:

- 1) The analysis of the nascent RNA-seq data in Figure 6D and E is confusing and as far as I understand inconsistent with the H3K4me3 data and the conclusions drawn by

the authors. They do not support a global effect on transcription. The figure itself lacks a clear description of what data are shown and what the red and blue data represent (the scale is unlabeled and goes from -2 to 2). It says Z-score in the legend, and based on the method description I am guessing that it represents statistically significant differential transcript levels between the short and long washout. But if these are significant transcript differences, why are the control data shown and why were the replicates processed separately? In any case, it looks like the authors only show differentially expressed transcripts, which means that the effect is not global and that they are probably documenting a specific cellular response to the cAMP treatment. It is well possible that the first cellular response includes transcription down-regulation, followed by an increase or a recovery from this effect. However, the authors need to show this better, with a direct comparison between ChIP-seq and nascent RNA-seq data. If the effect on transcription is more complicated, the authors need to tone down their conclusions regarding the regulation of gene expression.

2) The manuscript should be accessible to a general reader. While the overall message is clear, the section on how "cAMP signaling elevates labile RapGEF2-dependent V-ATPase assembly" could benefit from rewriting, having a more general audience in mind. Furthermore, it would be nice to include a model at the end of how the cAMP pathway affects H3K4me3 levels and a brief discussion on the general implications of why the cell might react to cAMP levels by decreasing H3K4me3 levels.

Minor points:

1) The concentration of cAMP used in their experiments is not consistent. In figure 1C, authors treat cells with 50uM of cAMP, while in figure S3, 100 uM of cAMP was used for the treatment. In figure S3, H34Me3 marks was completely abolished after 30 minutes treatment with 100 uM cAMP, while in figure 1C, 30 minutes of treatment with 50 uM cAMP (0 hours washout after cAMP treatment) did not abolish H3K4Me3. The authors should justify the use of different cAMP concentration in these experiments.

2) The authors observed that even in A2058 melanoma cells, elevated cAMP lead to a rapid reduction in H3K4Me3 mark, stating that this is a 'general effect'. Do the authors mean that the changes in H3K4Me3 levels induced by cAMP are general for those cells that express the specific Jmjc-domain containing histone methyltransferase? Or does cAMP induce similar effects on cells with low or no expression of this Jmjc-domain containing histone methyltransferase? This should be more clearly explained, and unless tested on further cell lines, I would recommend rewording to "likely a more general effect".

3) Changes in H3K4Me3 upon cAMP treatment in A2058 cells is not as strong as in Schwann cells. Does the cAMP treatment in A2058 cells lead to changes in H3K4Me2 or H3K4Me1 marks?

4) In Figure S5, what was the time of the CGPR treatment? We see a complete loss of H3K4Me3 after treatment with 10 nM CGPR in figure S5 (not sure about the time of treatment), but such an effect is not observed in figure 2A with 10 nM CGPR treatment.

5) Show examples of a few individual genes supporting changes in ChIP-seq and RNA-seq data, perhaps as a supplementary figure.

6) Figure S4 legend - typing error (H3K36Me3)

Reviewer #3 (Comments to the Authors (Required)):

Huff and collaborators provide insightful data on the role of G-protein coupled receptors and cAMP in the regulation of H3K4me3 levels. H3K4me3 is an important histone modification, associated mostly with transcriptionally active promoter regions. The authors demonstrate that cAMP increase correlates with decrease of H3K4me3, and the effect is reversed by "rescue" experiments with washout. They also suggest that cAMP-induced H3K4me3 demethylation is modulated by the co-factor Fe and that the demethylation of H3K4me3 in their system is regulated by KDM5 and HMTs. Finally they use genomics to demonstrate that the phenotypes they observe is also retrieved genome-wide on chromatin, with effects on nascent transcription. The manuscript is overall well written, the findings are novel and insightful. However, I believe there is room for improvement, as listed in the following points:

- 1) Abundant literature, has suggested that H3K4me3 is a co-transcriptionally deposited histone modification, meaning that it could be a consequence and not a determinant of active transcription. What are the authors thoughts about this? Is it possible that the effects on transcription that they show in Fig. 6 are actually directly due to cAMP elevation and that what they see on H3K4me3 is simply reflecting the transcriptional attenuation caused by cAMP. If the authors cannot demonstrate this (e.g. looking at levels of transcription factors associated to transcriptional initiation), they should at least discuss this point in the discussion.
 - 2) The authors never show p-values to support the significance of the genomic analyses showed in Fig. 6. While the decrease in H3K4me3 is visible, it is not clear to me if that is significant genome-wide.
 - 3) Nascent RNA analysis: how was differential gene expression measured on nascent RNA? Did the authors use only the exon to compute gene expression or the whole gene? In the latter case, how did the authors distinguished between gene transcription and transcription of intronic enhancers normally detected by nascent RNA techniques? This is one of the reasons why nascent RNA is normally not used to compute differential gene expression, but mostly to produce average gene profiles showing transcriptional levels. I would recommend the authors clarify on this.
-

Reviewer #1 (Comments to the Authors (Required)):

Main issue

1. How does cAMP addition affect cell proliferation and cycling? These may contribute to these results on histone methylation changes so this analysis should be performed.

Following the reviewer's suggestion, we performed both cell cycle and proliferation analyses of primary Schwann cells after cAMP treatment and washout. Using flow cytometry and propidium iodide staining to assess nuclear DNA content, we found that brief (30 minutes) treatment with cAMP or treatment followed by washout had no observable effect on cell cycle kinetics compared to untreated controls. No matter the condition, most Schwann cells (>90%) were observed in the G₀/G₁ phase while exhibiting negligible DNA synthesis (~0.3 - 0.5%), reflective of a primary cultured post-mitotic cell phenotype (Figure R1).

Figure R1. cAMP treatment and washout does not alter cell cycle kinetics. Flow cytometry analysis shows that >90% of Schwann cells are in the G₀/G₁ phase, ~5% in the G₂/M phase, and 0.3 - 0.5% in the S phase. The cell cycle is not altered by cAMP treatment (30 minutes, 50 μM) followed by washout.

To determine cell proliferation, we assessed the incorporation of EdU in primary Schwann cells. We found that neither brief cAMP treatment nor treatment followed by washout had any observable effects on EdU incorporation. In corroboration of the flow cytometry data, we found that EdU incorporation ranged between 0.1 - 0.2%, indicating nearly negligible proliferation in both control and treated conditions (Figure R2). Overall, the baseline proliferation rate of primary Schwann cells is low. Brief cAMP treatment followed by washout has no obvious impact on the cell cycle and proliferation of primary Schwann cells, suggesting the impact of cAMP on histone demethylation is less likely affected by cell cycle and proliferation changes.

Figure R2. cAMP treatment and washout does not alter Schwann cell proliferation. No significant change in EdU incorporation is found in Schwann cells after treatment with cAMP (30 minutes, 50 μ M) followed by washout. Scale bar = 20 μ m.

2. In Fig 1E and F, why are H3K4me3 levels increased after washout while diMe2 isn't, which is another substrate for KDM5 demethylases? Could there be additional effects on other enzymes that could explain these results?

After washout, the rebound of H3K4me3 but not H3K4me2 is likely to be explained by the dynamic methylation of H3K4, which is governed by the opposite activities of HMTs for methylation and (mainly) KDM5 for demethylation. H3K4 can be tri-, di-, mono-, or unmethylated (H3K4me3, H3K4me2, H3K4me1, and H3K4) respectively. H3K4me3 cannot be methylated further but can only be demethylated. Unmodified H3K4 cannot be demethylated further and can only be methylated. In contrast, both H3K4me2 and H3K4me1 can be methylated by HMTs and demethylated by KDM5, thus one may expect their state to be more transient relative to H3K4me3 or unmodified H3K4 (Figure R3). After washout of cAMP (removing the boost of KDM5 activity), HMT-mediated methylation of H3K4 accumulates at H3K4me3 but not H3K4me2 since the latter as a substrate could be further methylated to H3K4me3. To our understanding, this is similar to a cascade water fountain where the bottom basin, not middle ones, accumulates water. Overall, in our system cAMP-induced demethylation likely triggers a feedback mechanism leading to increased methylation following washout resulting in H3K4me3 accumulation.

Figure R3. Schematic of H3K4 methylation and demethylation. Histone methyltransferases (HMTs) establish H3K4 methylation while KDM5 demethylases antagonize H3K4 methylation.

3. This manuscript relies almost exclusively on inhibitors. Some genetic experiments should be performed to validate these results.

Following the reviewer's comments, we used siRNA to knockdown the expression of KDM5 in primary Schwann cells. The results show that cAMP treatment induced H3K4me3 demethylation in control cells and cells treated with non-targeting (Scramble) siRNA, while treatment with KDM5A/B/C siRNA abolished cAMP-induced H3K4me3 reduction (Figure R4). These data validate our finding with KDM5 inhibitor KDOAM-25, showing that cAMP-induced H3K4me3 demethylation is indeed mediated by KDM5 demethylases. We've now included these data in the supplementary materials (Figure S11) and have described these findings in the results section (Lines 256-260). Additionally, we

have updated the methods to describe our procedure for KDM5 siRNA silencing (Lines 529-537).

“We then used siRNA silencing of KDM5A/B/C isoforms to further validate their role in cAMP-induced H3K4me3 demethylation. cAMP caused significant H3K4me3 demethylation in Schwann cells provided no siRNA or treated with non-targeting (Scramble) siRNA. However, treatment with KDM5 siRNA abolished cAMP-induced H3K4me3 demethylation (Figure S11).”

Figure R4. KDM5 silencing abolishes cAMP-induced H3K4me3 demethylation.

Immunofluorescence and quantification show that while cAMP (50 μ M, 8 h) induces demethylation of H3K4me3 in non-siRNA and Scramble siRNA conditions, treatment with KDM5A/B/C siRNA abolishes cAMP-induced demethylation. * $P < 0.05$. Data represented are mean \pm SD. Scale bar = 20 μ m.

4. When genetic experiments were performed in Fig 4, H3K4me3 and Fe(II) levels were not even examined. Fig 4 appears to be a little peripheral to the main story so the experiments performed on H3K4me3 should be done for Fig 4D and 4E (4E lacks labels for the IF images as well).

We attempted to use the available RapGEF2 knockout NS-1 cells to examine the mechanistic role of RapGEF2 in labile Fe(II) elevation caused by cAMP. Following the reviewer’s suggestions, we assessed H3K4me3 after forskolin and isoproterenol treatment in both NS1 wildtype (WT) and NS1 RapGEF2 knockout (KO) cells. Our preliminary results show that treatment with either forskolin (100 μ M) or isoproterenol (10 μ M) had no drastic effect on H3K4me3 in both NS1 WT and RapGEF2 KO cells (Figure R5). This result is not surprising considering our working hypothesis regarding the mechanism underlying cAMP-induced H3K4me3 demethylation. We found that cAMP causes an increase in intracellular labile Fe(II) which serves as a cofactor for KDM enzymes that demethylate histones. According to our existing RNA-seq data, the highest expressed histone demethylases in primary Schwann cells are KDM5 enzymes which antagonize H3K4me3 (Table S1). Although H3K4me3 demethylation via KDM5 is most prominently observed in Schwann cells, we do not believe H3K4me3 alterations will be identified in all other cell types upon cAMP treatment using the relatively crude methods such as immunostaining and immunoblot, mainly because different cell types express different KDM enzymes which antagonize different histone methylation marks. Perhaps high-throughput methods such as ChIP-seq and quantitative mass spectrometry could identify the changes in histone methylation profile upon cAMP stimulation in NS-1 cells, which is out of the focus of this manuscript. Following the reviewer’s comments, we’ve included labels for the IF images in this panel.

Figure R5. cAMP signaling does not alter H3K4me3 in NS1 cells. Immunofluorescence shows no obvious H3K4me3 changes in NS1 wildtype (WT) and RapGEF2 knockout (KO) cells treated with forskolin (100 μ M) or isoproterenol (10 μ M). Scale bar = 20 μ m.

5. It is surprising that there are no changes in H3K4me3 levels in control and time 0 experiments when histone demethylase and methyltransferase inhibitors are used. How do the authors know that these are working?

We thank the reviewer's comments on inhibitors' effective time. In our experiment, time 0 represents the time at the moment of washout after cAMP treatment. Prior to cAMP treatment, cells were pretreated with HMT inhibitors (HMTi) MM-102 and MI-2. We chose the pretreatment time based on the time course effect of HMTi on H3K4me3 in Schwann cells. As shown in Figure R6, treatment with HMTi showed a drastic H3K4me3 decrease after 8 h but had no effect at 1 h or 3 h in Schwann cells. In our experiment in Fig. 5B, we pretreated the cells with HMTi for 3 h before treating with cAMP for 0.5 h followed by washout. At time 0, cells had been exposed to HMTi only for 3.5 h, which explains why H3K4me3 remained at a similar level as the control at this timepoint. We used HMTi in order to test if the rebound of H3K4me3 at 3.5 h time point after washout was mediated by HMT. Indeed, decreased H3K4me3 was observed at timepoint of 3.5 h after washout. In total, 7 h of exposure to HMTi could allow HMTi to effectively inhibit H3K4 methylation.

Figure R6. Time course treatment with HMT inhibitors. Immunofluorescence shows treatment with histone methyltransferase inhibitors MM-102 and MI-2 (20 μ M) decreases basal H3K4me3 only after 8 h of treatment. Scale bar = 20 μ m.

Following the reviewer's comments, we also conducted time course experiment with KDM5 inhibitor KDOAM-25, which showed an increase in H3K4me3 only after 48 h but not earlier (Figure R7). The original report of KDOAM-25 by Tumber et al. who synthesized the inhibitor (Tumber et al. 2017, *Cell Chem Biol* **24**: 371-380) also suggested 48 h treatment for KDM5 inhibition. In light of these new time course results and after consulting the original publication, we realized that the pretreatment of 24 h in our original experiment in Fig. 5A may not be sufficient for KDOAM-25 to effectively inhibit KDM5 activity. Subsequently, we repeated the experiment in Fig 5A but now pretreated cells for 48 h with KDOAM-25 (Figure R8). Cells without pretreatment expectedly showed a

decrease in H3K4me3 0.5 h after washout which rebounded after 3.5 h. When pretreated with KDOAM-25 for 48 h, cells at time 0 and control conditions exhibited higher levels of basal H3K4me3 compared to non-pretreated cells. Additionally, KDOAM-25 pretreatment abolished cAMP-induced H3K4me3 demethylation 0.5 h after washout. Figure 5 A and C in the manuscript have been updated by these new data. Again, we thank the reviewer's very helpful comments to improve this experiment.

Figure R7. Time course treatment with KDM5 inhibitor. Immunofluorescence shows treatment with pan-KDM5 inhibitor (100 μ m) increases basal H3K4me3 only after 48 h of treatment. Scale bar = 20 μ m.

Figure R8. KDM5 demethylases underlie H3K4me3 dynamics after cAMP treatment. Immunofluorescence and quantification of H3K4me3 show that brief treatment with cAMP followed by washout induces peak demethylation of H3K4me3 0.5 h after washout. Pretreatment with pan KDM5 demethylase inhibitor KDOAM-25 abolishes the effect of cAMP on H3K4me3. ** $P < 0.01$. *** $P < 0.001$. All data are means \pm SD. Scale bar = 20 μ m

6. Similarly, why is there no difference between control and 0 hr in Fig 2C while clearly in B there is a large difference in fluorescence intensity?

Cells were treated with CGRP (10 nM) for 0.5 h in Fig 2A and with LPA (2 μ M) for 1 h in Fig. 2B. In this case, different reagents, concentrations, treatment times, and receptors could underlie the upregulation of H3K4me3 immediately after the LPA treatment (time 0) but no obvious changes in H3K4me3 immediately after the CGRP treatment (time 0), compared to respective controls (baselines).

7. What type of genes are dysregulated by cAMP treatment in Fig 6? Are they related to cAMP and GPCR signaling? For the most highly altered H3K4me3 peaks and gene expression alterations, it would be nice to see some representative examples showing ChIP-Seq and RNA-Seq data.

To understand what types of genes were regulated by cAMP treatment, we performed Gene Ontology (GO) Biological Process pathway analysis. We found that cAMP treatment and washout 3.5 h upregulated pathways such as protein targeting to the endoplasmic reticulum and several pathways involving protein membrane targeting. Conversely, pathways such as positive chemotaxis and positive regulation of triglyceride biosynthesis were downregulated (Figure S12) as presented in our original manuscript. Following the reviewer's comments, we have now included representative examples of genes with corresponding changes in transcription and H3K4me3 in the supplemental materials. Table S4 shows the top 20 highest upregulated genes with increased H3K4me3 content 3.5 h following washout since a majority of differential transcripts were upregulated following washout.

Table S4: Top 20 upregulated genes with increased H3K4me3 content 3.5 h following washout.

Gene	Transcripts (RCPM average)		Fold change (0.5-3.5 h)	P-value	Direction	H3K4me3 (FPKM average)		Fold change (0.5-3.5 h)	P-value	Direction
	0.5 h	3.5 h				0.5 h	3.5 h			
Mycl	6.437	40.890	6.352	0.023855	Up	50.566	53.849	1.065	0.0356562	Up
Snai1	5.981	36.327	6.074	0.03543592	Up	14.648	16.916	1.155	9.9155E-08	Up
Nenf	12.418	66.621	5.365	0.00125936	Up	21.098	23.165	1.098	0.01166033	Up
Serpine1	154.108	777.558	5.046	1.91E-29	Up	4.093	7.713	1.884	5.6203E-05	Up
Serf1	15.180	73.453	4.839	0.00050988	Up	42.953	47.867	1.114	0.00061459	Up
Ckap4	36.341	162.440	4.470	3.88E-06	Up	35.554	39.030	1.098	0.00036628	Up
Ran	29.288	128.356	4.383	1.73E-05	Up	34.469	37.119	1.077	0.02649335	Up
Ranbp1	20.544	89.013	4.333	0.03218038	Up	38.061	40.499	1.064	0.02757395	Up
Calm3	150.729	641.432	4.256	3.47E-11	Up	49.190	52.965	1.077	0.01166152	Up
Tagln	51.682	216.917	4.197	2.93E-10	Up	8.457	11.375	1.345	0.00034476	Up
Nefl	63.483	260.502	4.103	1.15E-07	Up	47.803	52.169	1.091	0.00099953	Up
Rps4x	27.302	108.684	3.981	0.00034232	Up	21.719	24.398	1.123	0.02528571	Up
Cct6a	48.463	191.586	3.953	1.15E-07	Up	24.756	26.893	1.086	0.01893565	Up
Tprg1l	17.486	68.915	3.941	0.00168239	Up	23.679	25.224	1.065	0.04002287	Up
Map1a	171.003	665.936	3.894	3.97E-22	Up	24.501	27.164	1.109	0.01545091	Up
Slc9a3r2	22.073	84.797	3.842	0.01833788	Up	25.376	36.709	1.447	6.9403E-36	Up
Smim10l1	19.472	74.201	3.811	0.00586357	Up	27.309	30.287	1.109	0.00934728	Up
Gldn	75.766	283.216	3.738	1.96E-08	Up	21.309	29.424	1.381	2.9075E-27	Up
Pik3c2a	153.813	561.815	3.653	1.26E-16	Up	43.211	47.160	1.091	0.00715901	Up
Hnrnpa2b1	32.050	116.638	3.639	0.00158627	Up	37.418	40.915	1.093	0.005269	Up

Reviewer #2 (Comments to the Authors (Required)):

Major points:

1. The analysis of the nascent RNA-seq data in Figure 6D and E is confusing and as far as I understand inconsistent with the H3K4me3 data and the conclusions drawn by the authors. They do not support a global effect on transcription. The figure itself lacks a clear description of what data are shown and what the red and blue data represent (the scale is unlabeled and goes from -2 to 2). It says Z-score in the legend, and based on the method description I am guessing that it represents statistically significant differential transcript levels between the short and long washout. But if these are significant transcript differences, why are the control data shown and why were the replicates processed separately? In any case, it looks like the authors only show differentially expressed transcripts, which means that the effect is not global and that they are probably documenting a specific cellular response to the cAMP treatment. It is well possible that the first cellular response includes transcription down-regulation, followed by an increase or a recovery from this effect. However, the authors need to show this better, with a direct comparison between ChIP-seq and nascent RNA-seq data. If the effect on transcription is more complicated, the authors need to tone down their conclusions regarding the regulation of gene expression.

The heatmap presented displays RNA-seq expression data in accordance with the standards of the field (Hervera et al. 2018, *Nat Cell Biol* **20**: 307-319), and is intended to give the reader an overview of changes in the transcriptome. Each row of the heatmap shows a color representation of the Z-scores of expression levels for an individual transcript. Red colors represent high expression and blue values represent lower expression. The Z-scores cannot be generated without using the control samples as well, and control expression is traditionally included in heatmaps as well as individual replicates. Nondifferential transcripts are not shown in heatmaps as they are typically uninformative in comparison to differential transcripts following treatment. The term ‘global’ in our manuscript was used to indicate a large shift in the transcriptome, indicating that many genes change in expression. Our intention was not to mean that all transcripts in the genome change in expression following treatment, but rather that these changes are widespread. To avoid confusion and provide clarity, we’ve changed the word “global” to “widespread” in the sequencing section of the manuscript and the discussion. Additionally, the Figure 6D figure legend regarding the heatmap now states “Heatmap of the relative abundance of reads for statistically significant differential nascent transcripts after brief cAMP treatment followed by washout represented by plotted Z scores” to better explain the presented data.

2. The manuscript should be accessible to a general reader. While the overall message is clear, the section on how "cAMP signaling elevates labile RapGEF2-dependent V-ATPase assembly" could benefit from rewriting, having a more general audience in mind.

Following the reviewer’s suggestion, we have rewritten this section to make it clearer and more accessible to a general audience (Lines 208-244).

3. Furthermore, it would be nice to include a model at the end of how the cAMP pathway affects H3K4me3 levels and a brief discussion on the general implications of why the cell might react to cAMP levels by decreasing H3K4me3 levels.

Following the reviewer’s suggestion, we’ve now included a figure which summarizes our working model of GPCR-cAMP regulation of H3K4me3 demethylation as a new Figure 7 in the manuscript. Additionally, we’ve included a brief discussion on the general implications of this signaling pathway (Lines 371-378):

“Cells are constantly exposed to a barrage of extracellular stimuli and must dynamically respond to them in order to function and survive. cAMP-induced histone demethylation via GPCR activation could be a signaling mechanism employed by the cell in order to swiftly alter transcription in response to a dynamically changing microenvironment. Utilizing cAMP, the evolutionarily conserved second messenger, repurposes an intracellular process that is not only rapid but is responsive to a diversity of extracellular stimuli which activate GPCRs, thus conferring a more nuanced cellular response to dynamic environmental challenges.”

Figure 7. Working model of cAMP-induced H3K4me3 demethylation. Activation of Gs-coupled receptors leads to an increase in cAMP which results in RapGEF-mediated V-ATPase assembly, endosome acidification and subsequent labile Fe(II) release to the pool. Fe(II) activates KDM5 demethylases resulting in H3K4me3 demethylation. Conversely, stimulation of Gi-coupled receptors (not shown) will cause less labile Fe(II), suppressed KDM5 and higher H3K4me3 in Schwann cells.

Minor points:

1) The concentration of cAMP used in their experiments is not consistent. In figure 1C, authors treat cells with 50uM of cAMP, while in figure S3, 100 uM of cAMP was used for the treatment... The authors should justify the use of different cAMP concentration in these experiments.

Intracellular cAMP levels vary according to receptors and cell types, ranging from 10 nM to 10 - 50 μ M after stimulation of Gs-coupled receptors (Conti et al. 2014, *J Gen Physiol* **143**: 29-38). We used membrane-permeable 8-CPT-cAMP in our experiments. Since the cellular permeability of 8-CPT-cAMP is ~20% (Werner et al. 2011, *Naunyn Schmiedebergs Arch Pharmacol* **384**: 169-176), treatment with 100 μ M 8-CPT-cAMP can increase intracellular cAMP to ~20 μ M, which is within the endogenous range. We first observed cAMP-induced H3K4 demethylation using this dose. However, we found in later experiments that 50 μ M (~10 μ M intracellular concentration) was sufficient to induce H3K4 demethylation. To minimize potential off-target effects, we therefore continued our experiments using this dose. All main figures within the text (excluding the initial experiment Figure 1A) are performed using 50 μ M 8-CPT-cAMP.

...In figure S3, H3K4me3 marks was completely abolished after 30 minutes treatment with 100 uM cAMP, while in figure 1C, 30 minutes of treatment with 50 uM cAMP (0 h washout after cAMP treatment) did not abolish H3K4me3...

A higher concentration of cAMP likely induces a quicker response in H3K4me3 compared to a lower concentration due to the relative difference in cAMP availability within the medium following treatment.

2) The authors observed that even in A2058 melanoma cells, elevated cAMP lead to a rapid reduction in H3K4me3 mark, stating that this is a 'general effect'. Do the authors mean that the changes in H3K4me3 levels induced by cAMP are general for those cells that express the specific Jmjc-domain containing histone methyltransferase? Or does cAMP induce similar effects on cells with low or no expression of this Jmjc-domain containing histone methyltransferase? This should be more clearly explained, and unless tested on further cell lines, I would recommend rewording to "likely a more general effect".

We suggested that labile Fe(II) elevation and histone demethylation is likely a general effect of cAMP. However, the exact demethylation response would be contingent upon the expression of different KDMs in the cell. For primary Schwann cells, the highest expressed KDM enzymes are KDM5A, KDM5B, and KDM5C (Table S1) which demethylate H3K4me3. However, KDM expression profiles vary between cells and may therefore produce cell type dependent effects on histone demethylation in response to cAMP. We re-wrote the section regarding the response of A2058 cells to cAMP and how this is likely a general effect (Line 119-125).

“We also tested A2058 melanoma cells which, like Schwann cells, are of neural crest origin and may express similar Jmjc domain-containing histone demethylases and therefore exhibit H3K4me3 demethylation upon cAMP treatment. The rapid reduction in H3K4me3 was also observed in A2058 cells along with a swift recovery towards baseline levels following cAMP removal (Figure S4). Collectively, these results suggest that the regulation of histone demethylation could be a general effect of cAMP signaling”

We also further elaborate upon this result in the discussion (Lines 411-418):

“The regulation of histone demethylation could be a general effect of cAMP signaling, as the suppression of H3K4me3 by cAMP was also verified in A2058 melanoma cells. A2058 cells, like Schwann cells and melanocytes, are of neural crest origin and may share similar Jmjc domain-containing histone demethylase expression profiles, which could explain why H3K4me3 is preferentially demethylated in both cell types. Since all Jmjc domain-containing histone demethylases utilize Fe(II) for demethylase activity, it is expected that cAMP signaling would fluctuate different histone marks depending on the Jmjc histone demethylase expression profile of a given cell type”

3) Changes in H3K4me3 upon cAMP treatment in A2058 cells is not as strong as in Schwann cells. Does the cAMP treatment in A2058 cells lead to changes in H3K4me2 or H3K4me1 marks?

Indeed, cAMP treatment only caused a relatively smaller H3K4me3 change in A2058 cells. Following the reviewer's suggestion, we assessed changes in H3K4me2/1 after cAMP treatment followed by washout in A2058 cells. Although we observed a trend of H3K4me2 decrease and a trend of H3K4me1 increase 0.5 h after washout, there was no

statistically significant difference in H3K4me2 or H3K4me1 following cAMP (30 minutes, 50 μ M) treatment and washout (Figure R9).

Figure R9. cAMP treatment followed by washout does not affect H3K4me2/1 in A2058 melanoma cells. Immunoblot of H3K4me2 and H3K4me1 in A2058 melanoma cells after cAMP (50 μ M) treatment followed by washout. Band density quantifications show no significant change in H3K4me2 and H3K4me1 after cAMP treatment followed by washout.

4) In Figure S5, what was the time of the CGPR treatment? We see a complete loss of H3K4Me3 after treatment with 10 nM CGPR in figure S5 (not sure about the time of treatment), but such an effect is not observed in figure 2A with 10 nM CGPR treatment.

The treatment time for Figure S5 was 2 h, whereas the treatment time for Figure 2A was 0.5 h followed by washout. This may explain why an effect by CGRP at 10 nM is observed in Figure S5 but not in Figure 2A. We've now amended the figure legend for Figure S5 to include the treatment time.

5) Show examples of a few individual genes supporting changes in ChIP-seq and RNA-seq data, perhaps as a supplementary figure.

We have now included Table S4 that shows the top 20 transcription upregulated genes with increased H3K4me3 content 3.5 h following washout in the supplemental materials.

6) Figure S4 legend - typing error (H3K36Me3)

This has now been corrected.

Reviewer #3 (Comments to the Authors (Required)):

1) Abundant literature, has suggested that H3K4me3 is a co-transcriptionally deposited histone modification, meaning that it could be a consequence and not a determinant of active transcription. What are the authors thoughts about this? Is it possible that the effects on transcription that they show in Fig. 6 are actually directly due to cAMP elevation and that what they see on H3K4me3 is simply reflecting the transcriptional attenuation caused by cAMP. If the authors cannot demonstrate this

(e.g. looking at levels of transcription factors associated to transcriptional initiation), they should at least discuss this point in the discussion.

H3K4me3 is traditionally associated with the promoters of transcriptionally active genes. Whether H3K4me3 alone is a consequence or a determinant, both, or neither, of active transcription is an ongoing inquiry in the field. Following the reviewer's suggestion, we investigated the expression of general transcription factors (i.e. TFIIB, TFIID, etc.), which regulate transcription initiation in Schwann cells, from our nascent RNA-seq data (Sainsbury et al. 2015, *Nature Reviews Molecular Cell Biology* **16**: 129-143). No general transcription factor assessed exhibited changes in nascent transcription after cAMP treatment followed by 0.5 h or 3.5 h washout. However, from our data it is difficult to include or exclude the potential influence of co-transcriptional H3K4me3 deposition. We therefore add the following sentence in the discussion (Lines 389-392)

“H3K4me3 is known to be co-transcriptionally deposited. Although H3K4me3 is classically associated with the promoters of transcriptionally active genes, it remains unclear whether cAMP-induced H3K4me3 is a consequence or a determinant of the differentially transcribed genes.”

Additionally, we examined our RNA-seq data for transcription factors listed in the TRANSFAC transcription factor database which showed 11 transcription factors were upregulated and 9 downregulated from 0.5 - 3.5 h following washout (Table R1). At this time, it is difficult to include or exclude the potential influence of these transcription factors on the observed increase in transcription 3.5 h following washout.

Table R1: Transcription factor expression in Schwann cells from 0.5 hrs following washout to 3.5 hrs

Gene	Transcripts (RCPM average)		Fold change (0.5-3.5 h)	P-value	Direction
	0.5 h	3.5 h			
Hoxd8	1.072837958	26.13017574	24.35612532	0.005940516	Up
Snai1	5.980846525	36.32654469	6.07381322	0.035435919	Up
Atf3	99.36823947	348.7149622	3.509320121	7.87E-08	Up
Jun	171.7550545	601.931899	3.504594963	2.45E-17	Up
Nr2f2	92.93121172	233.5979252	2.513664902	0.000244784	Up
Nr2f1	45.08443955	110.9270518	2.460428763	0.015308977	Up
Jund	69.46400687	158.6498607	2.283914618	0.023500601	Up
Sox2	105.9657427	230.2078702	2.172474465	0.009433013	Up
Foxk2	66.40596851	140.4734865	2.11537441	0.028531816	Up
Nfe2l1	73.45965299	153.6861492	2.09211646	0.015611338	Up
Nfat5	108.5928452	210.4578394	1.938045172	0.011446495	Up
Meis2	357.3470176	211.605327	0.59215641	0.014286092	Down
Pax7	170.5469864	86.29192101	0.505971538	0.027918401	Down
Prdm1	187.5771635	86.89188612	0.462166527	0.009534017	Down
Pou2f3	162.1247582	65.47301422	0.403843404	0.011777765	Down
Hnf4g	103.2034101	34.83149971	0.33750338	0.001247075	Down
Pax2	146.1674191	46.17535623	0.31590731	0.000359888	Down
Twist2	76.38246115	22.71391687	0.297370843	0.013854358	Down
Mlxipl	41.24926894	10.59633403	0.256885378	0.021106751	Down
Pitx2	26.06906201	2.268771303	0.087029265	0.011234797	Down

2) The authors never show p-values to support the significance of the genomic analyses showed in Fig. 6. While the decrease in H3K4me3 is visible, it is not clear to me if that is significant genome-wide.

Following the reviewer's comment, we re-examined our ChIP-seq data using edgeR. Overall, 16,364 total peaks were analyzed. From control to the 0.5 h, 1,391 peaks were downregulated and 856 peaks were upregulated. From the 0.5 h to 3.5 h, 2,364 peaks were downregulated and 2,277 peaks were upregulated. These results showed that we indeed detected a global net decrease in H3K4me3 at 0.5 h following washout as observed using other methods (immunoblot, immunofluorescence). However, we did not observe a significantly higher net recovery in H3K4me3 at 3.5 h. Although the trend of H3K4me3 recovery is obvious if all peaks are considered, lack of statistical significance is likely due to sample variability at this time point and smaller replicates. Considering our findings obtained from orthogonal approaches, we consider these results to be biologically significant. We have now included this differential analysis as part of the main text (Line 273-278). Additionally, we've amended the Figure 6A legend to make clear that the heatmap presented shows all peaks rather than differential peaks only.

“Differential peak analysis showed that 1,391 peaks were downregulated and 856 peaks were upregulated at 0.5 h, resulting in an observed net decrease in H3K4me3. Although a trend was observed, differential peak analysis found that 2,364 peaks were downregulated and 2,277 peaks were upregulated at 3.5 h, resulting in no net increase at 3.5 h likely due to the high variability between samples in this condition.”

3) Nascent RNA analysis: how was differential gene expression measured on nascent RNA? Did the authors use only the exon to compute gene expression or the whole gene? In the latter case, how did the authors distinguished between gene transcription and transcription of intronic enhancers normally detected by nascent RNA techniques? This is one of the reasons why nascent RNA is normally not used to compute differential gene expression, but mostly to produce average gene profiles showing transcriptional levels. I would recommend the authors clarify on this.

In our method, we computed expression using the whole gene and the program edgeR, a statistical package designed for analyzing differential expression using count data from sequencing experiments. In our experiment we captured nascent transcripts which have yet to undergo intronic splicing. Due to the nature of nascent reads tending to contain these pre-processed transcripts, reads that contained exonic segments were all included whether or not part of the read aligned into an intron. The purpose of our nascent RNA seq analysis was to show that average gene transcriptional profiles were changing following rapid cAMP stimulus which co-occurred with changes in H3K4me3 and would potentially not be captured using other RNA seq methods due to the short nature of our treatment. To make this point clear, we have included this explanation within the methods of the manuscript under the “Nascent RNA-seq” section (Lines 584-586).

“Due to the nature of nascent reads to contain pre-processed transcripts which have not undergone RNA splicing, reads containing both exonic and intron segments were aligned and included in our analysis.”

In conclusion, we thank all the reviewers for their many helpful comments and suggestions that have led to a significant improvement of our revised manuscript. We would also like to thank you for your time and effort in handling our manuscript. While few limitations remain, we feel that the results of our study are timely and will be of interest to the wider research community working on cAMP signaling, Fe(II) regulation, and histone demethylation.

December 8, 2019

Re: Life Science Alliance manuscript #LSA-2019-00529-TR

Dr. Gaofeng Wang
University of Miami
1501 NW 10th Avenue
Miami 33136

Dear Dr. Wang,

Thank you for submitting your manuscript entitled "Oscillatory cAMP Signaling Rapidly Alters H3K4 Methylation and Gene Expression" to Life Science Alliance. The manuscript was assessed by the original reviewers again, whose comments are appended to this letter.

As you will see, the reviewers appreciate the introduced changes but still raise some concerns that need to get addressed. Doing so should be feasible in a further minor revision and following the constructive input provided by the reviewers. We would thus like to invite you to submit a final version of your manuscript to us. Importantly:

- Please address the remaining reviewer concerns
- Please add a callout to figure S12
- Please upload Table S2 and S3 as separate files.
- Please provide short titles for the tables (in the manuscript docx file)
- Please mention in the figure legends which statistical test has been used whenever mentioning p-values
- Any duplications across several figures need to get explained within the manuscript text
- Please remove the legends from all figures (also from suppl figure files), all legends, including S figure legends, should be in the main manuscript docx file

Thank you for this interesting contribution to Life Science Alliance. We are looking forward to receiving your revised manuscript.

Sincerely,

Andrea Leibfried, PhD
Executive Editor
Life Science Alliance
Meyershofstr. 1
69117 Heidelberg, Germany
t +49 6221 8891 502
e a.leibfried@life-science-alliance.org
www.life-science-alliance.org

B. MANUSCRIPT ORGANIZATION AND FORMATTING:

Reviewer #1 (Comments to the Authors (Required)):

The authors have improved the manuscript with some new data and answers to reviewer's comments that were provided. Overall, the manuscript and its reported

findings are better developed and the main scientific message that cAMP impacts histone methylation is more supported by data. However, this reviewer would still like the authors to resolve these two points that were not adequately addressed in their revision. These are important questions that upon answering fully will make this study appropriate for publication in LSA.

There are still a few issues that need to be addressed.

1. While the data is consistent with KDM5A, KDM5B and/or KDM5C being involved in cAMP-induced H3K4me3 demethylation, these experiments lack essential controls. The authors need to show that the siRNAs are knocking down these proteins and it would be preferable to do this separately, as well as in combination, to be able to say if there is demethylase that is more important than another and if it is really true that all 3 can compensate for each other. As is, the presented result in Figure S11 is not performed with the right controls to draw any strong and scientifically rigorous conclusions.

2. It was asked by multiple reviewers that some representative examples showing ChIP-Seq and RNA-Seq were provided (for example genes observed in UCSD browser with +/- treatment and H3K4me3 reads and for RNA-Seq, the reads of these genes. It may be that the authors misinterpreted this request as they did indicate that they showed some representative examples of genes and their changes in transcription after treatments but this is not what was requested. These additional data are required to better visualize and support the findings presented in Fig 6. For H3K4me3 and RNA-Seq, it may be better to show Control and 0.5, as well as 3.5. For Table S4, while the top 20 upregulated genes exhibit large fold-change expression, the fold change for H3K4me3 appears to be modest, what are the top 20 genes for fold change for H3K4me3 and how does the transcription of these genes change when one uses H3K4me3 demethylation as a marker to identify these genes. This is important given the focus of H3K4me3 demethylation by cAMP in this study. Given the large decrease in H3K4me3 levels by IF after cAMP treatment and washout (Fig 1C), this is difficult to understand given the modest changes in H3K4me3 that are shown using genomic analysis. This may just be how the data is presented but as a reader, this is a distraction so it should be explained in the text and some of these other analyses provided to better understand the data and its meaning.

Reviewer #2 (Comments to the Authors (Required)):

The revised manuscript by Huff et al is significantly improved and overall is of broad interest. However, my concerns related to the RNA-seq experiments and analysis in Figure 6 have not been properly addressed.

1) The authors did not make any direct comparison of the ChIP-seq and RNA-seq data as requested. Without this comparison, it is difficult to assess whether there is a correlation between cAMP-induced changes in H3K4Me3 and nascent transcription. A similar concern was also raised by reviewer 3 (point 1), questioning whether there is a causal relationship.

2) The analysis and presentation of the RNA-seq data in Figure 6D is still not appropriate. The authors used a standard R package, edgeR, but the way the analysis is performed and shown is not standard. Normally, replicates are used to assess the technical variation in order to assess whether differences between biological samples are significant. From the plot, it looks like replicates were often very inconsistent. This could either mean that the observed differences are very small and noisy (a Z-score does not say anything about the fold-change), or it could mean that the replicates were not processed properly. This is probably why Figure 6E looks so confusing: why are there so many up-regulated transcripts between the control and 0.5 h treatment, if

H3K4me3 is down-regulated? Based on the variation shown in Figure 6D, this might simply be noise.

3) Even if the data were better analyzed and presented, I am still doubtful that the data support the authors' claim. The fact that any changes in transcription are observed is not surprising. Given that only a fraction of genes is differentially regulated at $FDR < 0.05$ and that these changes are likely small (no fold-change is shown), claiming that there are widespread changes is not supported. And even if the changes were widespread, these changes could be secondary effects. Therefore, unless there really is a correlation between H3K4me3 levels and transcription, the RNA-seq data do not add anything to the manuscript.

Given these issues, I recommend removing the RNA-seq analysis from the manuscript.

Reviewer #3 (Comments to the Authors (Required)):

The authors addressed my concerns. I am still skeptical about using nascent RNA to perform differential gene expression, but I believe this should not prevent the paper from being accepted.

Reviewer #1 (Comments to the Authors (Required)):

1. While the data is consistent with KDM5A, KDM5B and/or KDM5C being involved in cAMP-induced H3K4me3 demethylation, these experiments lack essential controls. The authors need to show that the siRNAs are knocking down these proteins and it would be preferable to do this separately, as well as in combination, to be able to say if there is demethylase that is more important than another and if it is really true that all 3 can compensate for each other. As is, the presented result in Figure S11 is not performed with the right controls to draw any strong and scientifically rigorous conclusions.

Following the reviewer's suggestion, we assessed knockdown of KDM5A, KDM5B, and KDM5C from this experiment using qPCR. qPCR analysis shows that Schwann cells treated with KDM5 siRNA showed significant decrease of KDM5A and KDM5B compared to non-targeting Scramble siRNA following cAMP treatment (Figure R1). KDM5C was not detected. In this experiment, KDM5 siRNA treatment included a combination of siRNAs targeting KDM5A, KDM5B, and KDM5C in order to determine if KDM5 mediates cAMP-induced H3K4me3 demethylation. It is currently unknown whether KDM5 family members can compensate for each other, though we feel this is outside the scope of this manuscript. We believe that siRNA silencing of KDM5 members in conjunction with our presented data using the pan KDM5 inhibitor support the conclusion that KDM5 enzymes underlie cAMP-induced H3K4me3 demethylation.

Figure R1. Knockdown efficiency of KDM5A and KDM5B. qPCR analysis shows that Schwann cells treated with KDM5 siRNA showed significant knockdown of KDM5A (**A**) and KDM5B (**B**) compared to non-targeting Scramble siRNA following cAMP treatment. * $P < 0.05$. Data represented are mean \pm SEM

2. It was asked by multiple reviewers that some representative examples showing ChIP-Seq and RNA-Seq were provided (for example genes observed in UCSD browser with +/- treatment and H3K4me3 reads and for RNA-Seq, the reads of these genes. It may be that the authors misinterpreted this request as they did indicate that they showed some representative examples of genes and their changes in transcription after treatments but this is not what was requested. These additional data are required to better visualize and support the findings presented in Fig 6. For H3K4me3 and RNA-Seq, it may be better to show Control and 0.5, as well as 3.5.

Indeed, the previous request was misinterpreted. Due to the comments of Reviewer 2 and others, we have now removed nascent RNA-seq analysis from this manuscript. Therefore, only changes in H3K4me3 are presented. Following the suggestion of this reviewer, we have now created visualizations of H3K4me3 peak changes follow cAMP treatment in *Samd1* and *Fgpt*, two representative example genes from our ChIP-seq data using Integrative Genomics Viewer (IGV) (Figure R2). These data have now been included in the main figures as Figure 6D.

Figure R2. Visualizing H3K4me3 following cAMP treatment. Representative H3K4me3 peaks from *Samd1* and *Fgpt* which exhibited H3K4me3 demethylation 0.5 h after washout and recovery of H3K4me3 following 3.5 h. Plots were created using Integrative Genomics Viewer (IGV). Peak coordinates are indicated in grey.

For Table S4, while the top 20 upregulated genes exhibit large fold-change expression, the fold change for H3K4me3 appears to be modest, what are the top 20 genes for fold change for H3K4me3 and how does the transcription of these genes change when one uses H3K4me3 demethylation as a marker to identify these genes.

Since the RNA-seq analysis has been removed from the manuscript, now only H3K4me3 changes are considered. Following the reviewer's suggestion, we have now created a table of the top 20 genes showing largest fold decrease from Control to 0.5 hr (Table S4). Additionally, we have created another table showing the top 20 genes with the largest fold increase from 0.5 hr to 3.5 hr (Table S5). These have now been included as supplementary tables.

Table S4: Top 20 genes with decreased H3K4me3 content 0.5 hr following washout.

Gene	H3K4me3 (FPKM average)		Fold change (Control-0.5 hr)	P-value	Direction
	Control	0.5 hr			
Eln	15.500	0.426	0.027	1.2737E-14	Down
LOC684762	29.021	21.572	0.743	9.20006E-08	Down
Samd1	15.722	12.567	0.799	0.000315187	Down
Pdss1	24.739	19.853	0.802	2.2416E-05	Down
Gmcl1	21.561	17.530	0.813	0.000117551	Down
Zfp692	37.185	30.486	0.820	1.91061E-13	Down
Stk11ip	31.900	26.186	0.821	3.29336E-09	Down
Tmem164	23.676	19.865	0.839	9.80615E-05	Down
Hist1h3b	37.236	31.454	0.845	0.000764667	Down
Hist1h1d	31.596	27.017	0.855	6.34875E-07	Down
Slc25a23	24.590	21.061	0.856	0.009436625	Down
Tshz2	28.529	24.605	0.862	5.3357E-05	Down
Ptges3	48.359	41.919	0.867	3.39016E-13	Down
Cd99	41.013	35.660	0.869	1.17628E-08	Down
AABR07045680.1	41.013	35.660	0.869	1.17628E-08	Down
Ctgf	35.508	31.028	0.874	0.000527055	Down
Glis2	32.634	28.602	0.876	1.9859E-07	Down
Trim68	33.613	29.611	0.881	0.00456025	Down
Fpgt	28.214	24.865	0.881	0.000463449	Down
Lrriq3	28.214	24.865	0.881	0.000463449	Down

Table S5: Top 20 genes with increased H3K4me3 content 3.5 hr following washout.

Gene	H3K4me3 (FPKM average)		Fold change (0.5-3.5 hr)	P-value	Direction
	0.5 hr	3.5 hr			
Eln	0.426	8.268	19.419	0.002288761	Up
Slc25a23	21.061	28.564	1.356	4.3313E-06	Up
LOC684762	21.572	29.169	1.352	3.103E-05	Up
Samd1	12.567	16.306	1.297	0.001046237	Up
Gmcl1	17.530	22.066	1.259	0.00125468	Up
Vdac3	24.754	31.026	1.253	9.37995E-12	Up
Tmem185b	30.271	37.441	1.237	2.16811E-12	Up
Kank2	18.563	22.934	1.235	4.65794E-17	Up
Trim68	29.611	36.577	1.235	7.03004E-05	Up
Lrrc4	31.507	38.674	1.227	1.10305E-08	Up
AABR07044049.1	13.552	16.505	1.218	3.26731E-11	Up
Fpgt	24.865	30.238	1.216	9.71468E-07	Up
Lrriq3	24.865	30.238	1.216	9.71468E-07	Up
Gorasp2	41.535	50.240	1.210	0.000576718	Up
Mir3065	26.182	31.588	1.206	2.20583E-15	Up
Dlx2	39.460	47.599	1.206	4.63459E-18	Up
Ptp4a1	27.589	33.193	1.203	0.000208515	Up
Mtcp1	29.590	35.403	1.196	3.39171E-07	Up
Lmln	24.439	29.229	1.196	1.68122E-05	Up
Pdss1	19.853	23.710	1.194	0.018169733	Up

Given the large decrease in H3K4me3 levels by IF after cAMP treatment and washout (Fig 1C), this is difficult to understand given the modest changes in H3K4me3 that are shown using genomic analysis. This may just be how the data is presented but as a reader, this is a distraction so it should be explained in the text and some of these other analyses provided to better understand the data and its meaning.

Indeed, changes in H3K4me3 peaks detected by ChIP-seq were relatively modest compared to results obtained via IF or western blot for overall H3K4me3. It is not exactly clear to us why only modest changes were discovered in the ChIP-seq analysis. Two trained bioinformaticians in our paper (Sant and Van Booven) suggest that very high percentage of reads reside within large peaks, which may consequently affect the read coverage in other relatively smaller peaks that could be altered by the treatment. This read coverage bias toward large peaks may explain the relatively modest changes by ChIP-seq analysis. Following the reviewer's suggestion, we have put this explanation within the discussion (Lines 364-370).

Reviewer #2 (Comments to the Authors (Required)):

1) The authors did not make any direct comparison of the ChIP-seq and RNA-seq data as requested. Without this comparison, it is difficult to assess whether there is a correlation between cAMP-induced changes in H3K4Me3 and nascent transcription. A similar concern was also raised by reviewer 3 (point 1), questioning whether there is a causal relationship.

2) The analysis and presentation of the RNA-seq data in Figure 6D is still not appropriate. The authors used a standard R package, edgeR, but the way the analysis is performed and shown is not standard. Normally, replicates are used to assess the technical variation in order to assess whether differences between biological samples are significant. From the plot, it looks like replicates were often very inconsistent. This could either mean that the observed differences are very small and noisy (a Z-score does not say anything about the fold-change), or it could mean that the replicates were not processed properly. This is probably why Figure 6E looks so confusing: why are there so many up-regulated transcripts between the control and 0.5 h treatment, if H3K4me3 is down-regulated? Based on the variation shown in Figure 6D, this might simply be noise.

3) Even if the data were better analyzed and presented, I am still doubtful that the data support the authors' claim. The fact that any changes in transcription are observed is not surprising. Given that only a fraction of genes is differentially regulated at $FDR < 0.05$ and that these changes are likely small (no fold-change is shown), claiming that there are widespread changes is not supported. And even if the changes were widespread, these changes could be secondary effects. Therefore, unless there really is a correlation between H3K4me3 levels and transcription, the RNA-seq data do not add anything to the manuscript.

Given these issues, I recommend removing the RNA-seq analysis from the manuscript.

We thank the reviewer for their comments regarding our nascent RNA-seq analysis. After careful consideration, we agree with the reviewer that the RNA-seq analysis adds little overall to the message of the manuscript. Due to the concerns of this reviewer and others, we have followed the reviewer's suggestion and have removed the RNA-seq analysis from the manuscript. The title of the manuscript now reads: "Oscillatory cAMP Signaling Rapidly Alters H3K4 Methylation".

Reviewer #3 (Comments to the Authors (Required)):

The authors addressed my concerns. I am still skeptical about using nascent RNA to perform differential gene expression, but I believe this should not prevent the paper from being accepted.

Due to the concerns of this reviewer and others, the nascent RNA-seq analysis has been removed from the manuscript. We thank the reviewer for their support of our manuscript.

Editor's requests

Please address the remaining reviewer concerns

We have addressed the remaining concerns from the reviewers.

Please add a callout to figure S12

Following the suggestions of Reviewer 2 and others, we have removed the nascent RNA-seq analysis from the manuscript. Therefore, Figure S12 has been removed.

Please upload Table S2 and S3 as separate files.

Table S2 and S3 have been made and uploaded as separate files.

Please provide short titles for the tables (in the manuscript docx file)

Table titles have now been added to the manuscript docx file.

Please mention in the figure legends which statistical test has been used whenever mentioning p-values

Figure legends now mention the statistical test that was used to generate p-values.

Any duplications across several figures need to get explained within the manuscript text

As addressed in our previous correspondence, we have now removed unintended duplicated controls from Figure 4E/S10 which were performed in the same experiment and Figures S3/S8 which were also performed in the same experiment. We have now provided figures with different control images from their respective experiments. We thank the editors for noticing this unintended error.

Please remove the legends from all figures (also from suppl figure files), all legends, including S figure legends, should be in the main manuscript docx file

Legends from all figures have been removed and have now been placed in the main manuscript docx file.

In conclusion, we thank all the reviewers for their many helpful comments and suggestions that have led to a significant improvement of our revised manuscript. We would also like to thank you for your time and effort in handling our manuscript. While few limitations remain, we feel that the results of our study are timely and will be of interest to the wider research community working on cAMP signaling, Fe(II) regulation, and histone demethylation.

December 16, 2019

RE: Life Science Alliance Manuscript #LSA-2019-00529-TRRR

Dr. Gaofeng Wang
University of Miami
1501 NW 10th Avenue
Miami 33136

Dear Dr. Wang,

Thank you for submitting your Research Article entitled "Oscillatory cAMP Signaling Rapidly Alters H3K4 Methylation". It is a pleasure to let you know that your manuscript is now accepted for publication in Life Science Alliance. Congratulations on this interesting work.

DISTRIBUTION OF MATERIALS:

Again, congratulations on a very nice paper. I hope you found the review process to be constructive and are pleased with how the manuscript was handled editorially. We look forward to future exciting submissions from your lab.

Sincerely,

Andrea Leibfried, PhD
Executive Editor
Life Science Alliance
Meyerhofstr. 1

69117 Heidelberg, Germany
t +49 6221 8891 502
e a.leibfried@life-science-alliance.org
www.life-science-alliance.org